# Aerosol size distribution changes in FIREX-AQ biomass burning plumes: the impact of plume concentration on coagulation and OA condensation/evaporation

Nicole A. June[1], Anna L. Hodshire[2], Elizabeth B. Wiggins[3], Edward L. Winstead[3,4], Claire E. Robinson[3,4], K. Lee Thornhill[3,4], Kevin J. Sanchez[3], Richard H. Moore[3], Demetrios Pagonis[5,6†], Hongyu Guo[5], Pedro Campuzano-Jost[5], Jose L. Jimenez[5,6], Matthew M. Coggon[5,9], Jonathan M. Dean-Day[7], T. Paul Bui[8], Jeff Peischl[5,9], Robert J. Yokelson[10], Matthew J. Alvarado[11], Sonia M. Kreidenwis[1], Shantanu H. Jathar[12], Jeffrey R. Pierce[1]

[1]Department of Atmospheric Science, Colorado State University, Fort Collins, CO, USA
[2]Handix Scientific, Fort Collins, CO, USA
[3]NASA Langley Research Center, Hampton, VA, USA
[4]Science Systems and Applications, Hampton, VA, USA
[5]Cooperative Institute for Research in Environmental Sciences, University of Colorado Boulder, Boulder, CO, USA
[6]Department of Chemistry, University of Colorado, Boulder, CO, USA
[7]Bay Area Environmental Research Institute, Moffett Field, CA, USA
[8]Atmospheric Science Branch, NASA Ames Research Center, Moffett Field, CA, USA
[9]NOAA Chemical Science Laboratory (CSL), Boulder, CO, USA
[10]Department of Chemistry, University of Montana, Missoula, MT, USA
[11]Verisk Atmospheric and Environmental Research, Lexington, MA, USA
[12]Department of Mechanical Engineering, Colorado State University, Fort Collins, CO, USA

[†]Now at: Department of Chemistry, Weber State University, Ogden, UT, USA

*Correspondence to*: Nicole A. June (nicole.june@colostate.edu), Jeffrey R. Pierce (jeffrey.pierce@colostate.edu)

**Abstract.** The evolution of organic aerosol (OA) and aerosol size distributions within smoke plumes are uncertain due to the variability in rates of coagulation and OA condensation/evaporation between different smoke plumes and at different locations within a single plume. We use aircraft data from the FIREX-AQ campaign to evaluate differences in evolving aerosol size distributions, OA, and oxygen to carbon ratios (O:C) between and within smoke plumes during the first several hours of aging as a function of smoke concentration. The observations show that the median particle diameter increases faster in smoke of a higher initial OA concentration (>1000 µg m$^{-3}$) with diameter growth of over 100 nm in 8 hours–despite generally having a net decrease in OA enhancement ratios–than smoke of a lower initial OA concentration (<100 µg m$^{-3}$), which had net increases in OA. Observations of OA and O:C suggest that evaporation and/or secondary OA formation was greater in less concentrated smoke prior to the first measurement (5–57 minutes after emission). We simulate the size changes due to coagulation and dilution and adjust for OA condensation/evaporation based on the observed changes in OA. We found that coagulation explains the majority of the diameter growth with OA evaporation/condensation having a relatively minor impact. We found that mixing between the core and edges of the plume generally occurred on timescales of hours, slow enough to maintain differences in aging between core-edge, but too fast to ignore the role of mixing for most of our cases.

# 1 Introduction

Open biomass burning (landscape fires, including wildfires) is a significant source of aerosols and vapors in the atmosphere (Akagi et al., 2011; Gilman et al., 2015; Hatch et al., 2015; Jen et al., 2019; Reid et al., 2005; Yokelson et al., 2009). Aerosol particles emitted through biomass burning are composed almost entirely of organic compounds (often >90% by mass), with additional minor contributions from black carbon (BC) and inorganic salts (Bond et al., 2013; Capes et al., 2008; Carrico et al., 2008; Cubison et al., 2011; Garofalo et al., 2019; Hecobian et al., 2011; Mardi et al., 2018; Reid et al., 2005). These aerosol particles impact the health and welfare of communities exposed to the smoke as well as the Earth's radiative budget and climate (Carrico et al., 2008; Ford et al., 2018; Gan et al., 2017; Liu et al., 2015; O'Dell et al., 2019; Petters et al., 2009; Ramnarine et al., 2019; Reid et al., 2016; Shrivastava et al., 2017). Smoke particles have a direct radiative effect by scattering/absorbing solar radiation (Alonso-Blanco et al., 2014; Charlson et al., 1991; Haywood and Boucher, 2000; Jacobson, 2001; Ramnarine et al., 2019) and an indirect effect on climate through acting as cloud condensation nuclei (CCN) that modify the cloud albedo and lifetime (Albrecht, 1989; Lee et al., 2013; Pierce and Adams, 2007; Ramnarine et al., 2019; Spracklen et al., 2011; Twomey, 1974).

Particle size and composition influence how aerosols impact the magnitude of the direct and indirect radiative effects and where aerosols deposit in humans, therefore impacting health (Kodros et al., 2018; Lee et al., 2013; Seinfeld and Pandis, 2016; Spracklen et al., 2011). Particles are deposited into different locations in the respiratory tract based on particle size, where smaller particles are more harmful because they can make it deep into the lungs (Hinds, 1999; Kodros et al., 2018), and the toxicity of particulate matter from wildfires has also been linked to particle size (Jalava et al., 2006; Johnston et al., 2019; Leonard et al., 2007). The absorption/scattering efficiencies of the aerosols are determined by their size and composition (Seinfeld and Pandis, 2016). The scattering and Angstrom exponents of biomass burning smoke are dependent on aerosol size and composition (Junghenn Noyes et al., 2020; Kleinman et al., 2020). The ability of aerosols to act as CCN and then impact cloud properties is determined by the particle diameter and hygroscopicity (Lee et al., 2013; Petters and Kreidenweis, 2007; Spracklen et al., 2011). Lee et al. (2013) found that CCN concentrations were highly sensitive to uncertainties in biomass burning diameter, and Ramnarine et al. (2019) showed both the aerosol indirect effect and the direct radiative effect of biomass burning were sensitive to the aerosol size. Therefore, to accurately determine the climate and health effects of biomass burning aerosols, the particle size distribution and its evolution must be well understood.

Aerosol number size distributions from biomass burning evolve after emission with size distributions tending to shift to larger sizes and to decrease in modal width due to condensation/evaporation and coagulation (Capes et al., 2008; Carrico et al., 2016; Hodshire et al., 2019b, 2021; Janhäll et al., 2010; Levin et al., 2010; Sakamoto et al., 2015, 2016). Janhäll et al. (2010) showed that fresh smoke (< 1 hour) had median diameters ranging from 100 nm to 150 nm with modal widths varying between 1.6 and 1.9, while aged smoke (several hours to several days) had larger median diameters ranging from 200 nm to 300 nm with modal widths of 1.3 to 1.6. The Biomass Burning Observation Project (BBOP) campaign observed particle diameters to statistically increase with aging with smoke sampled ~15 minutes after emission having median

diameters of 40 nm to 150 nm, and smoke with an age of ~3 hours having median diameters of 175 nm to 260 nm (Hodshire et al., 2021). Observations of regional haze dominated by smoke over Brazil were also observed to have an increase in particle diameter (120 nm to 180 nm) and a decrease in modal width (1.73 to 1.63) as it aged (Reid et al., 1998). Past modeling work has suggested the size distribution changes observed in biomass burning plumes are due to both condensation/evaporation and coagulation (Hodshire et al., 2019b; Sakamoto et al., 2016). Both of these studies estimated that coagulation had the largest effect on diameter changes at high concentrations with slow dilution rates. In the work of Hodshire et al. (2019b), the simulated diameter change due to both organic condensation and coagulation seen in four hours ranged from 10 nm in dilute plumes ($\Delta$OA less than 10 $\mu$g m$^{-3}$) to 125 nm in concentrated plumes ($\Delta$OA of 500 $\mu$g m$^{-3}$).

Coagulation reduces particle number, shifts the distribution to larger sizes, and narrows the modal width of the size distribution (Hodshire et al., 2019b; Janhäll et al., 2010; Sakamoto et al., 2016; Seinfeld and Pandis, 2016). The coagulation rate is proportional to the square of the number concentration (when the sizes are held fixed), meaning that more concentrated smoke plumes have more rapid growth due to coagulation. Hence, the initial concentrations in the plume affect the coagulation rate; and because dilution of relatively cleaner, background air into smoke plumes lowers number concentrations, the plume dilution rate also impacts the coagulation rate (Sakamoto et al., 2016).

Importantly, most chemical transport and climate models are too spatially coarse to resolve individual plumes and their dilution. In these models, the emissions are instantly diluted within the coarse gridboxes (10s of kilometers), thus underestimating the role of coagulation. To remedy this, Sakamoto et al. (2016) developed a parameterization of coagulation within sub-grid scale diluting smoke plumes. Ramnarine et al. (2019) used this sub-grid parameterization of biomass burning and found that representing this in-plume coagulation impacts the radiative effect of biomass burning, increasing the direct radiative effect by up to 4% and decreasing the indirect effect by 43%, underscoring the importance of near-source, sub-grid coagulation in shaping the aerosol size distribution and radiative effects.

Organic aerosol (OA) condensation/evaporation can also lead to growth/shrinkage of the median diameter (Hodshire et al., 2019b; Riipinen et al., 2011; Zhang et al., 2012). If there is secondary organic aerosol (SOA) formation in the smoke plume, this SOA can condense onto existing particles leading to growth of the size distribution; this has been suggested by lab studies of biomass burning aerosol and in past field campaigns (Bian et al., 2017; Cubison et al., 2011; Hodshire et al., 2019b; Reid et al., 1998; Yokelson et al., 2009). A substantial fraction of primary organic aerosol (POA) in biomass burning plumes is semi-volatile, allowing for POA evaporation from particles as the plume dilutes and cleaner air is entrained into the plume (Bian et al., 2017; Cubison et al., 2011; Huffman et al., 2009; Jolleys et al., 2015; May et al., 2015, 2013). Hence, similar to coagulation, the initial concentration and dilution rate influences the evaporation of POA in the plume. This evaporation acts to decrease particle size. The net change in OA in the smoke plume determines the overall impact of OA condensation/evaporation on the aerosol size.

Field observations have shown that OA enhancement ratios can increase, decrease, or remain constant in the first 24 h of physical smoke aging (Akagi et al., 2012; Hecobian et al., 2011; Hobbs et al., 2003; Jolleys et al., 2015; May et al., 2015; Sakamoto et al., 2015; Vakkari et al., 2014; Yokelson et al., 2009; Zhou et al., 2017). OA enhancement ratios are the in-

plume OA with the background (out-of-plume) concentration of OA removed (that is, the "background corrected" OA) normalized by an inert species, typically background corrected CO (Akagi et al., 2012); OA enhancement ratios correct for dilution, and show the net change in OA as the smoke ages. Some prior works suggest SOA condensation and POA evaporation are simultaneously occurring in smoke plumes with the balance between the two determining how net OA changes (Bian et al., 2017; Hodshire et al., 2019b, a; May et al., 2015; Palm et al., 2020). In addition to dilution-driven evaporation of the POA, OA enhancement ratios may decrease through temperature increases in the smoke plume (Selimovic et al., 2019, 2020). Akherati et al. (2022) performed OA simulations of wildfire plumes measured during the WE-CAN field campaign, which support this condensation-evaporation balancing hypotheses, showing that dilution-driven evaporation of POA and simultaneous production of SOA explains the lack of change in OA enhancement ratios often observed in field campaigns during the first 2 to 8 hours of physical aging. Theoretical work has shown that OA enhancement ratio and composition changes may also be related to plume concentration (Bian et al., 2017; Hodshire et al., 2019b). However, Hodshire et al. (2021) found no statistically significant relationship between OA enhancement ratio changes and smoke age or initial plume concentration with BBOP data.

As the smoke plume ages, OA also undergoes changes in composition. Oxygen to Carbon (O:C) elemental ratios of OA have been used as a tracer for oxidative aging and SOA in the smoke plumes. Field and lab campaigns have shown that O:C typically increases as the smoke plume ages (DeCarlo et al., 2008; Hodshire et al., 2019a, 2021). The O:C increases observed in smoke plumes help to explain the lack of observed change in OA enhancement ratio. The condensed SOA and the remaining POA have higher O:C than the evaporated POA, so as SOA increases and POA decreases, the overall O:C increases (Akherati et al., 2022; Hodshire et al., 2021, 2019a). POA evaporation from dilution is the controlling factor in the O:C increase (Akherati et al., 2022; May et al., 2015). In BBOP and WE-CAN, O:C increases were inversely related to OA concentrations measured at the first transects (Akherati et al., 2022; Hodshire et al., 2021). Often these first transects are at 15–30 minutes of smoke age, so OA enhancement ratio and O:C changes occurring prior to the first transect (due to SOA formation and POA evaporation) may also be important (Hodshire et al., 2019a). Therefore, since dilution to low concentrations drives the POA evaporation, plumes with lower concentrations at the first transect may have higher O:C and a lower OA enhancement ratio at the time of the first transect (Akherati et al., 2022).

As described above, the smoke concentrations (and subsequent dilution) influence the evolution of the smoke plume, including coagulation and OA evaporation/condensation rates. Smoke concentrations and dilution rates span orders of magnitude with plume size and atmospheric stability. Under the same atmospheric stability conditions, a larger plume will dilute more slowly than a smaller plume since it will take longer for the background air to mix into the core of the plume (Bian et al., 2017; Hodshire et al., 2019b). The variability in plume size can lead to differences in dilution rates and concentrations, which can subsequently lead to differences in size, number, and OA at the time of the first measurement and beyond. Since fires range in size, it is important to consider the initial plume concentrations and dilution rates in studies working to understand plume aging; however, studies that use field work to determine this relationship are limited.

140

In addition to concentrations and dilution rates varying due to plume size, concentrations also vary based on the radial position in the smoke plume (Decker et al., 2021; Hodshire et al., 2021; Peng et al., 2020; Wang et al., 2021), leading to differences in coagulation and OA evaporation/condensation between the edge and core of a plume (Hodshire et al., 2021). Although fires span orders of magnitude in size with a large number of fires burning an area less than 0.1 km$^2$, field campaigns tend to only sample fires this size and larger (Hodshire et al., 2019a). However, we may be able to segregate sampled plumes into relatively concentrated and dilute sections to gain a better understanding of how smaller undersampled plumes may evolve, based on the evolution of the less-concentrated plume edges (Hodshire et al., 2021). Hodshire et al. (2021) used this method to examine the relationship of the following individual variables with initial OA mass concentration and physical smoke age using data from the BBOP campaign: OA mass, OA oxidation state, aerosol diameter, and aerosol number concentration. The analyzed smoke plumes did show differences in plume edge and core evolution, with evidence of O:C changes occurring rapidly prior to the first transect in less concentrated plumes and plume edges, and a correlation of diameter with plume age and concentration (Hodshire et al., 2021). However, the Hodshire et al. (2021) study did not consider mixing between radial portions of the plume within the smoke plume in their analysis, implicitly assuming that each more and less concentrated region evolved independently. They noted the need for improvement in understanding O:C and particle diameter changes based on initial plume concentrations as well as fuel type (Hodshire et al., 2021).

In this work, we use the observations of plumes in the western US during the FIREX-AQ campaign to examine the role of smoke concentration on variability in aerosol size and OA evolution between and within smoke plumes. Further, we evaluate the roles of coagulation and condensation/evaporation in the aerosol size changes. To help elucidate the role of smoke concentration on biomass burning aerosol size and OA evolution, we analyze the evolution of both transect-averaged smoke aerosol properties as well as the differences between the dilute and concentrated portions of the smoke plume. We use an aerosol-microphysics model to estimate how much of the aerosol size growth is due to coagulation versus OA condensation/evaporation; the first study to show in multiple Pseudo-lagrangian transects of smoke plumes the dominance of coagulation. Finally, we investigate the timescale of mixing between the more and less concentrated regions of plumes to determine if aging in these portions of the plumes can be assumed to occur independently; prior studies have not investigated this role of mixing. These analyses seek to parametrically link near-field smoke particle size distribution and composition properties at the time of the first transect to the subsequent evolution relevant for smoke in models. Thus, our findings should be of keen interest to the regional-to-global scale modeling community. In Sect. 2, we describe our methods. In Sect. 3, we first present our results based on the FIREX-AQ observations, then we present our results estimating the aerosol size changes due to coagulation and condensation/evaporation. We summarize our conclusions in Sect. 4.

## 2 Methods

### 2.1 DC-8 Aircraft Observation Data

The FIREX-AQ campaign took place in July–August 2019, sampling wildfire smoke in the western US and agricultural and prescribed fire smoke in the southeastern US. In our study, we use eight sets of transects from the NASA DC-8's deployment in the western US (Fig. 1), where the DC-8 aircraft crossed the plume repeatedly, generally moving from close to the fire to further downwind of the fire. The eight sets of transects are from four different fires on six days. The Williams Flats Fire was sampled twice on two of the days. The fuels burned varied among fires as well as between the different sampling days of the Williams Flats Fire (Table 1). The aircraft sampled free tropospheric smoke at altitudes varying from 2800 m to 5280 m above ground level, with temperatures varying from 267 K to 285 K.

Although a true Lagrangian sampling (sampling the same air parcel repeatedly over time as it moves downwind of the fire) is best for isolating the processes influencing aerosol aging, this is difficult to achieve. In FIREX-AQ, the DC-8 aircraft generally flew downwind at two to four times the wind speed at the sampling altitude (Table 1, Fig. S1), meaning that the smoke sampled farther from the fire had generally been emitted by the fire earlier in the day than the smoke sampled close to the fire. Due to this pseudo-Lagrangian sampling, observations can be impacted by the time-varying fire intensity (Wiggins et al., 2020). As a baseline test for the consistency in smoke emissions across the times where the sampled smoke was emitted, we excluded additional plume samplings from the western portion of the campaign due to those plumes having a non-zero slope ($p<0.05$) linear relationship between modified combustion efficiency (MCE) and plume age (Fig. S2). For the first set of transects of the Williams Flats Fire on 8/3 (Williams Flats 8/3 P1 on figures), transects are limited to those that are the most Lagrangian as identified by Wang et al. (2021). These were determined based on vertical locations within the plume from the LIDAR measurements. The transects not used in our analysis were towards the top of the plume, while the transects used in our analysis are vertically in the densest section of the plume (Wang et al., 2021). Additionally, the transect used to initialize the coagulation model (Sect. 2.2) is not the youngest smoke sampled in 6 cases under the constraint that the initialization transect should have the highest ΔCO (Table S2). The ages at the time of the first transect range from 5 to 57 min with most falling between 40 to 50 min. In our analysis, we assume that the changes in the smoke are due to physical aging; however, we expect that the deviation from perfectly Lagrangian sampling in the remaining sets of transects may still influence our results, and we discuss the implications of this potential influence throughout.

### 2.1.1 Aircraft Instruments

The TSI laser aerosol spectrometer (LAS) measured the particle size distribution between 0.1 and 5 μm at 1 Hz resolution. The LAS uses a helium-neon laser with the ability to detect particles as small as 90 nm in diameter and as large as 7.5 μm with 20% uncertainty across all sizes. The LAS was calibrated using size-classified ammonium sulfate aerosols (refractive index of $1.52 + 0i$), uncertainties exist in mass, volume, number, and size due to differences in the refractive index in the smoke aerosol (Moore et al., 2021). We apply corrections to the LAS measurements for both evaporation due to

heating in the sampling lines and optical saturation of the LAS sensor. Regarding saturation of the LAS measurements, we use work from Nault et al. (2018) to linearly extrapolate to higher aerosol number concentrations (from $2 \times 10^3$ cm$^{-3}$ to $2.3 \times 10^5$ cm$^{-3}$) to correct for this saturation after accounting for the LAS instrument dilution employed during FIREX-AQ. Although it is well known that the LAS saturates at high concentrations (which motivated the use of the dilution system), the functional dependence of this is unknown; therefore, there are some uncertainties introduced by assuming a linear dependence (Fig. S3), and we investigate this by examining the differences in our model simulations of median diameter when using a linearly extrapolated correction, a quadratically extrapolated correction, or no saturation correction (Nault et al., 2018). Next, we apply an evaporation correction for evaporation in the inlet tube due to temperature differences with the ambient air; evaporation due to the dilution system is not included. The evaporation correction is applied to the median particle diameters calculated from these size distributions based on calculations of the mass fraction remaining (MFR). The MFR is unique for each pseudo-Lagrangian set based on the ambient, inlet, and total temperatures; inlet pressure; and OA concentration. In the flights used in our analysis the ambient and inlet temperatures were typically 273 K and 300 K, respectively (Cappa, 2010; Pagonis et al., 2021). We assume that the fractional change in diameter from the evaporation correction is size independent and is found from the following equation

$$D_p = D_{p,measured} \left( \frac{1}{MFR} \right)^{\frac{1}{3}} \tag{1}$$

Figure S4a shows this evaporation correction for OA concentrations 1 µg m$^{-3}$ to 2000 µg m$^{-3}$ assuming a particle diameter of 300 nm for an ambient temperature of 273 K, inlet temperature of 300 K, and a pressure of 700 mb. Figure S4b and S4c show the impact on MFR and the diameter correction for an OA concentration of 1000 µg m$^{-3}$. To test the sensitivity of our results to the MFR being calculated for a 300 nm particle, the MFR is adjusted to be that of a particle equal to the median particle diameter based on the slope of Fig. S4b.

The Aerodyne high-resolution time-of-flight aerosol mass spectrometer (AMS) measured OA at 1 or 5 Hz time resolution (Canagaratna et al., 2007; DeCarlo et al., 2006; Guo et al., 2021; Nault et al., 2018; Xu et al., 2018). The uncertainty for the AMS OA has been estimated to be +/- 38% ($2\sigma$) mostly due to the uncertainties in the collection efficiency (CE) and the relative ionization efficiency of OA (RIE$_{OA}$) (Bahreini et al., 2009; Guo et al., 2021; Xu et al., 2018). CE was estimated according to the Middlebrook et al. (2012) composition-dependent algorithm (Middlebrook et al., 2012). A constant RIE$_{OA}$ of 1.4 was assumed for ambient particles based on previous studies (e.g. Jimenez et al., 2016; Xu et al., 2018) and calibrated pre-campaign with organic surrogates in the laboratory (Pagonis et al., 2021). As discussed in Guo et al. (2021), the AMS inlet had near 100% transmission between 70 and 635 nm vacuum aerodynamic diameter, equivalent to roughly 70 and 590 nm in (dry) aerodynamic diameter hence capturing the full accumulation distribution for typical FIREX-AQ plumes (Moore et al., 2021). We also applied the Pagonis et al., (2021) evaporation correction to the AMS data. However, the inlet residence time for the AMS, 0.3-0.4 s up to 8 km, was much shorter than that of the LAS, so the AMS MFR is generally much closer to 1 than that of the LAS (less evaporation for the AMS).

Regarding other DC-8 instruments used in this study, CO was measured by the NOAA LGR at 1 Hz resolution
(Bourgeois et al., 2022). The instrument operated with 2% uncertainty during the FIREX-AQ campaign. The meteorological measurement system (MMS) provides measurements of the 3D wind field, temperature, and turbulent dissipation rate. For the MMS we used 20 Hz measurements, instead of 1 Hz, to have a higher temporal resolution for calculating the turbulence.

### 2.1.2 Derived Parameters from Observations

The FIREX-AQ dataset provides background flags used for determining the background concentrations of species. Each fire sampled has a fire-ID in the dataset, which indicates when the DC-8 was sampling in a plume. The background concentrations for CO for the transects used in our analysis ranged from 76 to 166 ppb, with the minimum in-plume CO concentrations ranging from 98 to 300 ppb. The smoke age was provided in the dataset based on the aircraft-measured wind speeds and straight-line horizontal advection between the fire and aircraft position. As shown in Table 1, the aircraft moves downwind faster than this advection, so changes in emissions will affect the observations, and we note this as a limitation of our analyses. In addition to the uncertainties from the pseudo-Lagrangian sampling, there are likely uncertainties in the smoke age due to the wind shifting directions and potentially wind velocity varying radially within the plume (discussed more later in this section).

The concentration enhancement of species X due to the presence of smoke ($\Delta X$) is determined by subtracting the average background concentration ($X_{background}$) of this species from the in-plume measurements ($X_{inplume}$). Background concentrations are an average concentration measured outside the plume at the same altitude as the aircraft sampled the plume. We correct for dilution by creating an enhancement ratio (sometimes referred to as a normalized excess mixing ratio, NEMR; Akagi et al. (2012)). These enhancement ratios are found by normalizing the background-corrected species ($\Delta X$) by background-corrected CO ($\Delta CO$), since CO is inert on timescales of near-field aging (Yokelson et al., 2009)

$$\frac{\Delta X}{\Delta CO} = \frac{X_{inplume} - X_{background}}{CO_{inplume} - CO_{background}} \tag{2}$$

Increases or decreases in this enhancement ratio ($\Delta X/\Delta CO$) indicate production or removal of that species in the smoke plume (provided that the sampling is close enough to Lagrangian that variability in emissions do not impact changes in the observed enhancement ratios). In this study we look at $\Delta N/\Delta CO$ (number enhancement ratio), and $\Delta OA/\Delta CO$ (organic aerosol enhancement ratio, referred to as OAER).

Following Hodshire et al. (2021), mass concentrations of O and C are calculated using the AMS measurements of the O/C and H/C ratios. We assume that all OA mass is from O, H, and C, ignoring the contributions of Nitrogen and other minor organic atoms, allowing us to calculate background-corrected O/C using the following equation

$$\frac{\Delta O}{\Delta C} = \frac{\left(O_{inplume} - O_{background}\right)}{\left(C_{inplume} - C_{background}\right)} \tag{3}$$

The number median diameter ($D_{pm}$), number concentration (N) and modal width of the size distribution ($\sigma$) are calculated by fitting a lognormal distribution to the binned $dN/dlogD_p$ measurements from the LAS. N is the number

concentration between 50 nm and 2000 nm, the range of diameters used to fit the $dN/dlogD_p$ measurements. This size range, which extends slightly beyond the range of the LAS, allows us to see both the leading and trailing edges of the size distribution (Fig. S5). Based on the fits and LAS observations, we believe a single mode is enough to describe the size distribution. Additionally, Moore et al. (2021) did not show a smaller mode when the SMPS sampled in the smoke plume. We examine the change in number enhancement ratio within this size range.

For each of the variables described above: $D_{pm}$, $\Delta N/\Delta CO$, OAER, $\Delta O:\Delta C$, an ordinary least squares regression is used to calculate its average rate of change as the smoke ages. These rates of change are not intended to be extrapolated beyond 2 to 7 hours of aging. Our goal is to relate these average rates of change, as well as initial values of OAER and $\Delta O:\Delta C$ to the smoke concentration at the first transect, where smoke concentration is represented as the initial background-corrected organic aerosol concentration ($\Delta OA_i$). The average rates of change have uncertainty (95% confidence interval for the slope

of the ordinary least squares regression), which vary between sets of transects. To account for these varying uncertainties when determining the impact of initial smoke concentration on the rate of change of these variables, we use a Monte Carlo method to vary the rate of change of a data point within its 95% confidence interval assuming the data are normally distributed about the mean rate of change. For example, to determine the relationship between the rate of change of $D_{pm}$ ($dD_{pm}/dt$) and $log(\Delta OA_i)$. We perform 1000 Monte Carlo samples for each fit. The 95% confidence interval for the

relationship between $dD_{pm}/dt$ and $log(\Delta OA_i)$ is determined based on the 2.5 and 97.5 percentile of the slopes from the Monte Carlo linear regressions. We also perform the linear regressions assuming $dD_{pm}/dt$ to be the center of the 95% confidence interval, while sequentially removing one set of transects at a time. The Monte Carlo and the removing-one-set-at-a-time methods of fitting help to visualize and quantify the uncertainties of the relationship between the rate of change of each of our variables of interest and $log(\Delta OA_i)$. We use $\Delta OA_i$ as an indicator for the smoke concentration, and all fits with smoke

concentration are done on a linear-log scale, we have used $log(\Delta OA_i)$ since volatility distributions are thought of in orders of magnitude, and smoke concentration spans orders of magnitude.

In addition to using transect-average values, to investigate cross-plume gradients in evolution we divide each transect into $\Delta CO$ percentiles to evaluate the dilute and concentrated portions of the smoke plume separately. The percentiles used are 5 to 15, 15 to 50, 50 to 90, and 90 to 100, following (Hodshire et al., 2021). The lowest percentile bin starts at the 5th

percentile to provide a buffer between the background and in plume. The range of $\Delta CO$ values vary between transect (with smoke age) and between plumes; the average concentration of one plume may be similar to that in the edge of another. Figure S6 shows the locations of the percentiles in each of the eight plumes used in this analysis. We note that the spatial distribution of these percentiles within each smoke plume is complex, with the most concentrated percentiles not always falling in the physical center of the plume due to heterogeneous mixing with background air and a non-symmetric

distribution across the transect. The mixing may also mean that there are differences in the smoke age in the percentile bins due to the time for the initial momentum of the smoke plume to equilibrate with the velocity of the environmental air at the injection level. In Fig. S7, we show the ages of each percentile bin for each transect derived separately using the mean wind speeds in the percentile bins and the distance from the fire. While the derived ages vary by around 20 to 30 minutes between

the 5 to 15 and 90 to 100 percentile bins, there are no systematic differences with one bin being generally younger or older than the other. Further, the difference in the plane speed and wind speeds cause the imperfection in Lagrangian sampling to be larger than the variability in the smoke age in the percentiles. Therefore, we use the single value of smoke age for each transect included in the dataset for both the transect average and percentile bins.

The ability to gain insight into the differences in processes/aging between the dilute and concentrated portions of the same plumes may be limited if mixing between our CO-percentile regions is occurring on timescales faster than several hours (the aging time observed by the aircraft). We use the following procedure to estimate the timescale of this mixing within each plume. (1) The mean and standard deviations of each wind component are calculated using an averaging time approximately equal to the length of time the DC-8 spends sampling a plume transect. (2) The standard deviations of the cross-plume wind ($\sigma_v$) and vertical wind ($\sigma_w$) as well as the mean wind ($\bar{u}$) are used to approximate the lateral ($\sigma_\theta = \sigma_v/\bar{u}$) and vertical ($\sigma_\varphi = \sigma_w/\bar{u}$) turbulence intensities. (3) The Pasquill stability class (Arya, 1999) is estimated using these turbulence intensities (Table S1). (4) Gaussian dispersion relations are used to calculate a turbulent diffusivity, from which a mixing length is determined (Seinfeld and Pandis, 2016). (5) The distance and mixing time between the 5th to 15th percentile bin and the 90th to 100th percentile bin is calculated by using the geographic coordinates of the innermost point in the 5th to 15th percentile bin, and the average geographic coordinates of the 90th to 100th percentile bin. (6) The mixing length and distance between the percentiles is used to determine the mixing time. As a check on the mixing time calculated from the stability class, since we are extrapolating the Pasquill stability class to above the planetary boundary layer, we also calculate a mixing time from the rate of change of the $\Delta CO$ gradient between the core and edge regions. The $\Delta CO$-gradient derived mixing time is the inverse of

$$\frac{d((\Delta CO_{90-100} - \Delta CO_{5-15})/\Delta CO_{i,avg}}{dt} \tag{4}$$

where $\Delta CO_{90-100}$ ($\Delta CO_{5-15}$) is the $\Delta CO$ concentration in the 90-100 (5-15) $\Delta CO$ percentile bin, $\Delta CO_{i,avg}$ is the average $\Delta CO$ concentration at the first transect.

## 2.2 Coagulation Model

We use an aerosol microphysics box model to simulate the change in the aerosol size distribution due to coagulation and dilution in the smoke plumes. The model is initialized using the mean diameter, total number concentration, and the modal width of each smoke plume or $\Delta CO$ percentile based on a lognormal fit of the observed values at the first transect. These parameters are used to initialize the aerosol size distribution, which is represented with 1000 logarithmically spaced, single-moment size bins between 50 and 2000 nm. The model is run forward in time for 3 to 8 hours of aging depending on the maximum age of observations sampled in a particular set of transects. The model simulates Brownian coagulation using the Fuchs form of the Brownian coagulation kernel (Fuchs, 1964). In the Brownian coagulation kernel calculation, we assume a particle density of 1400 kg m$^{-3}$, and assume temperature and pressure are the average of the in-plume measurements.

Dilution is included in the model by using the observed first-order decay rate of $\Delta CO$. The dilution factor ($k_{dil}$) is used to calculate the rate of number change due to dilution in each size bin

$$\left(\frac{dN}{dt}\right)_{dil} = -N_{bin}k_{dil} \tag{5}$$

In the base simulations of this model, the aerosol size distribution is only changed at each time step through the combined effects of dilution and coagulation.

We show additional results, where the net evaporation and/or condensation of organic aerosol are also taken into account by using the observed linear fits of the $\Delta OA/\Delta CO$ ratio with smoke age for each set of transects. In this calculation, we assume that there is no new-particle formation, so all SOA condenses onto existing particles. Although new-particle formation may be occurring, particularly on the edges of plumes (Hodshire et al., 2021), these particles are too small to be measured by the instrumentation. Additionally, this assumes volume-controlled growth/shrinkage, where all particle sizes

grow/shrink by the same fractional amount, preserving the lognormal modal width. The modeled median diameter with the OA production/loss ($D_{pm,wOA}$) is included using the following equation

$$D_{pm,wOA} = D_{pm,coag}\left(\frac{d(\Delta OA/\Delta CO)}{dt}t + 1\right)^{\frac{1}{3}} \tag{6}$$

Where $\frac{d(\Delta OA/\Delta CO)}{dt}$ is the average observed change in the OA enhancement ratio with time from an ordinary least squares regression, and t is the simulation time. We assume that the evaporation and condensation does not impact the coagulation

rates, and is only an adjustment on the coagulation simulated median diameter ($D_{pm,coag}$). For small changes due to condensation/evaporation, the change in the modal width is small and it should not have a significant impact on the coagulation rate. For example, if there is less than a factor 2 change in OA mass, the change in the coagulation rate is less than 10% (Sakamoto et al., 2016; Seinfeld and Pandis, 2016). As we show in the results, the uncertainty due to these assumptions is smaller than the uncertainties in the measurements (e.g., saturation and evaporation corrections). Finally, we

acknowledge that diffusion-limited condensation, the Kelvin effect, and size-dependent differences in organic aerosol activity may lead to size-dependent growth/shrinkage differences, and this should be investigated in future work.

## 3 Results

### 3.1 Observations

      As shown in Fig. 2, all sets of transects have an increase in number-median transect-average diameter ($D_{pm}$) as the

355 smoke ages. Some flights have a consistent increase in $D_{pm}$ as the smoke ages, such as Williams Flats 8/3 P1 and Williams Flats 8/7 P2, while others have greater variability between each transect, such as Castle 8/12 and Williams Flats 8/6. Additionally, at the first transect the initial $D_{pm}$ varies from 150 nm to 225 nm, indicating potential differences in emissions and evolution prior to the first measurement (Fig. 2). Although we are performing a linear regression, we would expect the diameter growth rates to slow with age when growth is dominated by coagulation because coagulation rates slow as number

concentrations decrease from dilution and coagulation. The rate of the growth slow down varies between sets of transects, and in days such as Williams Flats 8/7 P2 is not noticable to slow dilution; the growth slow down is discussed more in Sect. 3.2 with the model results. Castle 8/12 also has a larger uncertainty in the linear fit, due to a constant increase in $D_{pm}$ for the first five hours of aging, but then a decrease in $D_{pm}$ during the final three transects, potentially due to deviation from Lagrangian sampling. This decrease does not appear to be due to particles smaller than 100 nm growing into the observed size range (Fig. S5). In the $\Delta CO$ percentiles within each plume, $D_{pm}$ also tends to vary at the first transect and increase with smoke age with varying degrees of uncertainty (Fig. S8). The other properties of the aerosol size distribution, modal width ($\sigma$) (Fig. S9) and $\Delta N/\Delta CO$ (Fig. S10) also have variability at the first transect and tend to decrease with smoke age.

Figure 3 shows that at the first transect (between 10-60 minutes after emission), each of the properties of the aerosol size distribution have a dependence on the initial smoke concentration ($\Delta OA_i$). Initial $D_{pm}$ and $\Delta OA_i$ have a Pearson correlation coefficient of 0.93 in the transect averages and 0.88 in the $\Delta CO$ percentiles. Based on these categories for a correlation coefficient: 0.0–0.19 is very weak, 0.2–0.39 is weak, 0.4–0.59 is moderate, 0.6–0.79 is strong and 0.8–1.0 is very strong (Evans, 1996), there is a very strong relationship between initial $D_{pm}$ and $\Delta OA_i$. As a function of $\Delta OA_i$, the initial $D_{pm}$ increases at a rate of 49.6 nm log($\mu$g m$^{-3}$)$^{-1}$ in the transect averages and 40.3 nm log($\mu$g m$^{-3}$)$^{-1}$ in the $\Delta CO$ percentiles (Fig. 3a-3b, Table S3). In our discussion of initial OAER, we will show that the lower initial $D_{pm}$ at the first transect can partially be explained by evaporation occurring prior to the first measurement. $\sigma$ and $\Delta N/\Delta CO$ at the first transect are also correlated to the smoke concentration. As $\Delta OA_i$ increases the initial $\sigma$ decreases by -0.06 log($\mu$g m$^{-3}$)$^{-1}$ in the transect averages with a Pearson correlation coefficient of -0.73 and by -0.05 log($\mu$g m$^{-3}$)$^{-1}$ with a Pearson correlation coefficient of -0.65 in the $\Delta CO$ percentiles. $\Delta N/\Delta CO$ at the first transect also significantly decreases as $\Delta OA_i$ increases with a rate of -40.3 cm$^{-3}$ log($\mu$g m$^{-3}$)$^{-1}$ in the transect averages and -17.7 cm$^{-3}$ log($\mu$g m$^{-3}$)$^{-1}$ in the $\Delta CO$ percentiles (Fig. 3e-3f). These differences seen in the properties of the aerosol size distribution at the first transect highlight the influence of processing through coagulation and evaporation occurring prior to the first measurement as well as how these process rates depend on plume concentration.

As initial smoke OA concentration increases, the average rate of increase of $D_{pm}$ increases, both for the transect averages and the $\Delta CO$ percentiles (Fig. 4a). As detailed in Sect. 2.1.2, to determine the relationship between initial smoke concentration ($\Delta OA_i$) and average rate of change of $D_{pm}$ while considering the uncertainty of these linear fits, we use a Monte Carlo method to vary the growth rate within the 95% confidence interval of each datapoint assuming the data are normally distributed about the mean for each datapoint. Using the Monte Carlo fitting method to consider these uncertainty ranges, the average rate of change of $D_{pm}$ with smoke age ($dD_{pm}/dt$) increases by 4.3 nm h$^{-1}$ log($\mu$g m$^{-3}$)$^{-1}$ with the 95% confidence intervals not crossing zero (Table S3), meaning that for every factor of 10 increase in initial OA concentration the growth rate increases by 4.3 nm h$^{-1}$. The use of $\Delta CO$ percentiles expands the range of concentrations and number of datapoints in determining the relationship between growth rate and initial smoke concentration (Fig. 4b), although mixing between percentiles may influence these trends, as discussed in Sect. 3.2. With the $\Delta CO$ percentiles, the Monte Carlo fitting has an average slope of 3.9 nm h$^{-1}$ for every factor of 10 increase in $\Delta OA_i$ and a reduction in the 95% confidence interval in comparison to the transect averaged relationship between $dD_{pm}/dt$ and $\Delta OA$ (Table S3). The Pearson correlation coefficient

of $dD_{pm}/dt$ and $\Delta OA_i$ is 0.53 and 0.43 for the transect averages and $\Delta CO$ percentiles, respectively. Similarly, the BBOP campaign showed moderate correlation between $D_{pm}$ and smoke age (Hodshire et al., 2021).

The width of the size distribution typically decreases with smoke age with an average Pearson correlation coefficient for all 8 sets of transects of -0.57 (Fig. S9). Additionally, the width decreases faster with smoke age as $\Delta OA_i$ increases with a slope of -0.01 $h^{-1}$ $\log(\mu g\ m^{-3})^{-1}$ in both the transect averages and $\Delta CO$ percentiles (Fig. 4c-d). This intuitively makes sense based on the hypothesis that coagulation is dominant in the smoke plumes and coagulation decreases the modal width in smoke plumes, since in more concentrated smoke the coagulation rate will be faster leading to a faster increase in $D_{pm}$ and a faster decrease in the width.

The aerosol number enhancement ratio is moderately correlated with smoke age with an average Spearman correlation coefficient for all 8 sets of transects of -0.73 (Fig. S10); while $D_{pm}$ with smoke age had a very strong relationship with an average Spearman correlation coefficient of 0.81. The aerosol number enhancement ratio could be less correlated with smoke age than $D_{pm}$ due to a changing N:CO emissions ratio from the fire during the period of imperfect pseudo-Lagrangian sampling (with the plane moving downwind ~4x faster than the wind speed). The BBOP campaign also showed the number enhancement ratio to have less of a relationship with smoke age than diameter (Hodshire et al., 2021). However, in Fig. S10, 5 of the 8 sets of transects have a tight correlation of number with age, and the high variability between transects for the remaining 3 sets of transects erode the average correlation, which may highlight the challenges of analyzing data that is not nearly Lagrangian. The Pearson correlation coefficients between the rate of change of number enhancement ratio and $\Delta OA_i$ are -0.46 in the transect averages and -0.55 in the $\Delta CO$ percentiles (Fig. 4e-f). Thus, this quantifier gives a moderate relationship between plume concentration and the rate of change of number enhancement ratio and growth rate of $D_{pm}$, which agrees with the results from Sakamoto et al. (2016) for plumes experiencing size distribution changes primarily through coagulation. Although the correlation coefficient for the transect averages gives a moderate relationship between the number enhancement ratio rate of change and $\Delta OA_i$ in the transect averages, taking into account the uncertainty of the rates of change in number enhancement ratio gives a non-statistically significant relationship with $\Delta OA_i$ of -2.2 $cm^{-3}$ $ppbv^{-1}$ $h^{-1}$ $\log(\mu g\ m^{-3})^{-1}$ (Fig. 4e, Table S3). The large 95% confidence interval in the transect averages is in part due to the high uncertainty of rate of change of number enhancement ratio in the Williams Flats 8/6 sampling because of variability in number enhancement ratio from transect to transect (Fig. S10). In the $\Delta CO$ percentiles, there is a statistically significant relationship between the rate of number enhancement ratio change with smoke age and $\Delta OA_i$ with an average trend of -4.4 $cm^{-3}$ $ppbv^{-1}$ $h^{-1}$ $\log(\mu g\ m^{-3})^{-1}$ (Fig. 4f, Table S3), although this may be influenced by mixing between percentiles that will be explored later.

The initial OAER ($\Delta OA/\Delta CO$) increases as the $\Delta OA_i$ increases (Fig. 5a). For the average values at the initial transect, this relationship has a slope of 0.17 $\mu g\ m^{-3}$ $ppbv^{-1}$ $\log(\mu g\ m^{-3})^{-1}$ with a p-value less than 0.01 and a Pearson's correlation coefficient of 0.91, with no apparent correlation to temperature (temperature should influence organic gas-particle partitioning). The lower OAER in dilute plumes ($\Delta OA_i$ less than 100 $\mu g\ m^{-3}$) suggests that there may be significant evaporation prior to the first transect; between the most concentrated plume (2085 $\mu g\ m^{-3}$) and the most dilute plume (45 $\mu g$

m$^{-3}$) around half of OA mass is lost, assuming no confounding SOA production (and no significant correlation between the OA/CO emissions ratio and the OA concentrations at the first transect). The lower initial $D_{pm}$ in dilute plumes (Fig. 3a) also suggests faster evaporation. Between a $\Delta OA_i$ of 100 µg m$^{-3}$ and 1000 µg m$^{-3}$, initial OAER decreases by a factor of about 0.62, which if only evaporation was occuring would suggest the particles in the 100 µg m$^{-3}$ plume would be 0.85 times smaller assuming the emitted diameter is not correlated with concentration at the first transect. We observe particles that are 0.76 times smaller at a $\Delta OA_i$ of 100 µg m$^{-3}$ compared to 1000 µg m$^{-3}$ (Fig. 3a), suggesting that evaporation prior to the first transect is contributing to smaller particle sizes for less-concentrated plumes. It is unlikely these differences are explained entirely by the variability in the age at the first transect. The smoke with the lowest $\Delta OA_i$ and initial OAER are the youngest at the first transect, meaning they have had less time for dilution and aging processes prior to this first measurement. The positive correlation between the initial OAER and $\Delta OA_i$ is consistent with WE-CAN observations and simulations done in Akherati et al. (2022); simulations of smoke plumes by Bian et al. (2017), Hodshire et al. (2019); and observations in Palm et al. (2020). A similar relationship is seen when binned by $\Delta CO$ percentiles, the initial OAER increases with increasing initial $\Delta OA$ at a rate of 0.12 µg m$^{-3}$ ppbv$^{-1}$ log(µg m$^{-3}$)$^{-1}$ with a p-value less than 0.01 and an R of 0.71 (Fig. 5b). There is also no correlation in this relationship to the average temperature at the first transect. Additionally, the initial OAER in the edges of the plume tends to be higher than that in the cores of the plume, suggesting that SOA production may be occurring more quickly at the edges (offsetting some evaporation) than at the core. However, there is no evidence of enhanced initial $\Delta O:\Delta C$ values at the edge over the core (Fig. 6b). We cannot rule out that this strong to very strong initial OAER trend with initial OA is also impacted by the burn conditions, although this would require the least concentrated smoke to have the lowest OA:CO emissions ratios rather than being controlled mostly by fire size, burn rates, and initial dilution. On the other hand, there is evidence that a significant fraction of smoke primary OA is semivolatile, such that we would expect evaporation of a fraction of this primary OA with dilution (May et al., 2013, 2015).

With aging, OAER either increases, decreases or remains about the same (Fig. 5c, 5d), with a moderate to strong negative correlation of dOAER/dt with increasing initial $\Delta OA$ (Pearson R of -0.62 and -0.51 in the transect averages and $\Delta CO$ percentiles). The average Monte Carlo slope is -0.03 µg m$^{-3}$ ppbv$^{-1}$ h$^{-1}$ log(µg m$^{-3}$)$^{-1}$ in the transect averages and -0.02 µg m$^{-3}$ ppbv$^{-1}$ h$^{-1}$ log(µg m$^{-3}$)$^{-1}$ in $\Delta CO$ percentiles (Fig. 5c,d); the 95% confidence intervals are in Table S3 and do not cross zero. There is some relationship with temperature in dOAER/dt, higher temperatures are more supportive of continued evaporation as the plume ages, while lower temperatures tend toward no net change or net condensation. This temperature correlation may be related to the effect of temperature on OA volatility as well as OA particle-phase diffusivity/viscosity (Maclean et al., 2021). The positive slopes seen at lower concentrations combined with the first transect being at least 30 minutes downwind (Fig. S11) is supported by prior theoretical work (Bian et al., 2017; Hodshire et al., 2019b). This prior work showed that for dilute plumes (those of initial OA concentrations less than 100 µg m$^{-3}$), there was an initial decrease in OAER followed by an increase in OAER starting after about 30 minutes (Bian et al., 2017; Hodshire et al., 2019b). Both the WE-CAN and BBOP campaign showed no significant change in OAER as the plumes aged, on average (Hodshire et al., 2021; Palm et al., 2020). Two samplings included here, Shady 7/25 and Williams Flats 8/7 P2, have no statistically

significant change in OAER as the smoke ages; OAER is variable between transects for Shady 7/25, however for Williams Flats 8/7 P2 OAER is consistent as the smoke ages (Fig. S11). Palm et al. (2020) showed that dilution driven evaporation of POA was a source of SOA in the fires, creating an overall balance in the OAER as the smoke aged, and this may be what is occurring in the Williams Flats 8/7 P2 sampling. The reduction of OAER seen at high concentrations was not observed in WE-CAN (Garofalo et al., 2019; Palm et al., 2020); however, the upper end of concentrations shown here for FIREX-AQ ($\Delta OA_i$ = 3000 µg m$^{-3}$) are greater than those from WE-CAN ($\Delta OA_i$ = 1700 µg m$^{-3}$). Although there is likely additional OA formation occurring in the concentrated plumes, it appears that dilution-driven evaporation of semi-volatile species dominates (Hodshire et al., 2019a; May et al., 2015). The OAER decrease with time in concentrated plumes may also be due to a slower rate of photochemistry in these concentrated plumes (May et al., 2013) since thick smoke plumes often have lower photolysis rates (Peng et al., 2020; Wang et al., 2021). Despite the decrease in OAER for concentrated smoke, which would act to decrease the particle diameter, the concentrated smoke still sees more growth (Fig. 3), which highlights the role of coagulation for growth and will be investigated further later.

The initial values of $\Delta O:\Delta C$ increase as plume concentration decreases with a very strong relationship (Fig. 6a-b). In the transect averages, this trend is -0.07 log(µg m$^{-3}$)$^{-1}$ (p-value< 0.01, R = 0.84) and in the $\Delta CO$ percentiles this trend is also -0.06 log(µg m$^{-3}$)$^{-1}$ (p-value< 0.01, R = 0.84). $\Delta O:\Delta C$ is higher in SOA than the evaporating POA (DeCarlo et al., 2008; Hodshire et al., 2019a, 2021). Additionally, the evaporating POA may have lower $\Delta O:\Delta C$ than the remaining POA that does not evaporate (Akherati et al., 2022), although the opposite trend has also been seen when only part of the POA to semi-volatile organic compound mass is captured (Jen et al., 2019). The increasing $\Delta O:\Delta C$ at the first transect as plume concentration decreases suggests that in dilute plumes there may be faster evaporation and/or SOA formation prior to the first transect. Higher initial $\Delta O:\Delta C$ in dilute plumes tends to have lower initial OAER (Fig. 5a-b); both indicate faster evaporation prior to the first transect in dilute plumes. There was evidence for this in the WE-CAN plumes as well (Akherati et al., 2022). In simulations of the WE-CAN plumes, Akherati et al. (2022) showed that it is likely that the POA evaporating prior to the first transect has a lower $\Delta O:\Delta C$, leaving the remaining POA with higher $\Delta O:\Delta C$. Further, Akherati et al. (2022) estimated that the more-dilute plumes contained a higher fraction of SOA at the first transect, further increasing the $\Delta O:\Delta C$ of the more-dilute plumes. Our results appear to be consistent with these findings of Akherati et al. (2022).

All plumes and $\Delta CO$ percentiles within plumes show a very strong increase in $\Delta O:\Delta C$ with smoke age with Spearman correlation coefficients of 0.93 in the transect averages and 0.96 in the $\Delta CO$ percentiles (Fig. S12), but there is no significant trend for the rate at which $\Delta O:\Delta C$ increases as the plume ages with the initial plume concentration ($\Delta OA_i$) in either case (Fig. 6c-d, Table S3). Therefore, the less-concentrated plumes and portions of plumes tend to continue to have higher $\Delta O:\Delta C$ ratios as the plume ages. Since the dilute plumes had a higher initial $\Delta O:\Delta C$, they continue to have higher $\Delta O:\Delta C$ values than the more-concentrated plumes at each plume age. The BBOP campaign had a moderate relationship of $\Delta O:\Delta C$ with smoke age (Hodshire et al., 2021). A review of published previous field campaigns (Hodshire et al., 2019a) shows that most field campaigns nearly always observe $\Delta O:\Delta C$ increasing with smoke age. Akherati et al. (2022) ran simulations for the WE-CAN campaign, which also observed increases in $\Delta O:\Delta C$ with smoke age. They found that dilution-driven evaporation of

semi-volatile POA played the strongest role in increasing $\Delta O:\Delta C$ (as opposed to SOA formation) (Akherati et al., 2022). This was likely because the lower-volatility organic compounds left in POA were similar to or higher in O:C than the additionally formed SOA. It is possible that this dilution-driven evaporation is what is dominating the $\Delta O:\Delta C$ increases and OAER decreases seen in the concentrated FIREX-AQ smoke plumes. These results suggest that concentration changes can be both cross plume and with time, while changes in composition are dependent on more than just oxidation and have a dependence on evaporation. In the cases where there is an increase or no change in OAER and an increase in $\Delta O:\Delta C$ with smoke age, there is likely both dilution-driven POA evaporation and significant SOA formation from semi-volatile organic compounds and volatile organic compounds.

## 3.2 Estimating the Drivers of the Observed Growth

Coagulation is the primary cause of growth in these smoke plumes with the rate being impacted by dilution, as shown by our simulations of coagulation and dilution in the plumes (the solid line in Fig. 7). With the exception of July 29 (Fig. 7a), coagulation explains the majority of the growth seen in the smoke plumes. For these days, the modeled coagulation often represented the growth of the median diameter within the uncertainty of the observed median diameter (Fig. 7). After the first transect (where the model and observation are forced to be equal), the modeled and observed median diameters have an average Pearson correlation coefficient of 0.82. Overall, across all cases, the mean absolute error after the first transect is 7 nm (mean bias -2 nm); however, this error is within the uncertainty range of the measurements. Additionally, coagulation alone does well at estimating the rate of change of the median diameter with smoke age, with a very strong Pearson correlation coefficient between the modeled $dD_{pm}/dt$ and the observed $dD_{pm}/dt$ of 0.8 (Fig. S13). Some of the disagreement between the model and observations may be due to imperfect Lagrangian aircraft sampling, especially noticeable in Fig. 7b and 7c, as discussed in Sect. 3.1. Non-Lagrangian sampling may be impacting the observations through a plume injection height change, so the aircraft is no longer sampling the same vertical location of the plume; or an emissions factor or fire radiative power change due to the diurnal cycle of fires. Our findings are supportive of estimations from Hodshire et al. (2021) and Sakamoto et al. (2016) that coagulation is the dominant process in changing the diameter in smoke plumes. The dilution rate also impacts the rates of the simulated $D_{pm}$ growth. Williams Flats 8/7 P2 is the slowest diluting plume with a dilution rate of 0.09 $h^{-1}$ with an average simulated growth rate of 19 nm $h^{-1}$; however, the Williams Flats 8/7 P1 simulation, which had a similar initial number concentration and modal width, diluted quicker at 0.43 $h^{-1}$ and only had an average simulated growth rate of 13 nm $h^{-1}$ due to a decreased growth rate after the first two hours. In both cases, the simulation accurately represents the observed growth rates of 20 nm $h^{-1}$ in Williams Flats 8/7 P2 and 14 nm $h^{-1}$ in P1, supportive of findings in Sakamoto et al. (2016) that a plume with a faster dilution rate has a slower coagulation rate due to the decrease in number concentration from dilution.

The agreement between modeled and observed $D_{pm}$ is potentially impacted by some of the assumptions that we made during our analysis, including assuming a linear function for the LAS saturation correction extension (Fig. S3) and assuming a non-size-dependent evaporation correction (i.e., all sizes have the same fractional size change due to evaporation; Fig. S4).

When we use no LAS saturation correction, the observed median diameter growth is underpredicted by the model (mean bias of -13 nm) (lower initial particle concentrations, so slower coagulation); on the other hand, when we use a quadratic function for the LAS saturation correction extension, the observed median diameter is overpredicted (mean bias of 12 nm) by the model (higher initial particle concentrations, so faster coagulation) (Fig. S14). Changing from the non-size-dependent evaporation correction to a size-dependent evaporation correction based on Fig. S4b does not change the agreement of the modeled and observed median diameters because the modeled coagulation rate is unchanged and only a minor shift in median diameter occurred because the assumed diameter of 300 nm used for the non-size-dependent evaporation correction is near the observed peak diameter (Fig. S15).

The observed trends in the number enhancement ratio are noisier than the trends in $D_{pm}$, but the model still is able to capture some of the reduction in number as a result of coagulation (Fig. S16). The average Spearman and Pearson correlation coefficients between modeled and observed number enhancement ratio are 0.57 and 0.52, respectively, a moderate relationship. These correlation values are negatively impacted by poor correlations between the model and observations on North Hills 7/29 and Williams Flats 8/6. The model did have a decrease in number enhancement ratio on both of these days; however, the noise in the observations, potentially due to experimental error from changing emissions, resulted in negative correlation between the modeled and observed number enhancement ratio (Pearson R of -0.14 for North Hills 7/29 and -0.65 for Williams Flats 8/6) (Fig. S16). The removal of these two samplings increases the average Pearson correlation coefficient to 0.83, a very strong relationship, similar to that between the modeled and observed $D_{pm}$. This result suggests the model is reasonably simulating the decrease in number enhancement ratio due to coagulation, but the model does not simulate number enhancement ratio as well as $D_{pm}$. As discussed with the differences in the observed trends, a possible explanation is the changing emissions due to the lack of Lagrangian sampling are impacting the number enhancements greater than they are impacting the diameter in the size distributions. 2

The model captures some of the reduction in the width of the size distribution (Figure S17). The average Spearman and Pearson correlation coefficients between the modeled and observed width are 0.52 and 0.50. Overall, as shown by the comparison of the normalized size distributions in Figure S18, the coagulation is reasonably able to simulate the observed changes in the size distribution.

In some cases, OA condensation/evaporation can further explain some of the growth (dashed lines in Fig. 7); however, this effect is often an adjustment that is smaller in magnitude than the variability of the $D_{pm}$ measurements. OA condensation/evaporation was included in the model based on the observed trends in OA (Fig. 5c) and Eq. 6. We are basing OA condensation/evaporation on the linear fit of the points in Figure S11, and there does not appear to be any systematic change in the slope as any of the plumes age. Consistent with the OAER trends (Fig. 5c), net condensation grows the particles in comparison to the coagulation-only model diameter in 3 cases, and net evaporation shrinks the model particle diameter in 5 cases (Fig. 7). The North Hills 7/29 case had the largest improvement as a result of including the observed condensation/evaporation effects. On this day, coagulation only increased the diameter by 5 nm, while coagulation and condensation combined increased the diameter by 15 nm, which was closer to the observed growth of 25 nm. Two cases,

Castle 8/12 and Williams Flats 8/3 P2, had reductions in model agreement with the inclusion of OA condensation/evaporation. For Castle 8/12, condensation resulted in an overestimation of the growth; however, this bias is not greater than the uncertainty of the measurement. Net evaporation was observed during Williams Flats 8/3 P2 resulting in underprediction of the growth. Overall, the changes due to including the OA condensation/evaporation were often small, and this is reflected in the mean absolute error only changing from 11 nm to 9 nm (Fig. 5). This relatively small change in model performance suggests that the condensation/evaporation had a minor effect on the changing median diameter in these plumes; however, due to variability between transects and uncertainties in the diameter measurements, it is unclear if including condensation/evaporation significantly improves the model. We also recognize that there is uncertainty in the role of condensation/evaporation due to the imperfect Lagrangian sampling of the plumes as well as uncertainties in the linear regressions of OAER vs. age. However, since some plumes were sampled more than once on the same day, and the times of day also varied, we think condensation/evaporation has a minor effect due to it not explaining a majority of the observed growth in any of the eight simulations. As discussed earlier, prior studies found that POA evaporation roughly balanced SOA formation, leading to no net change in OAER. In these cases condensation/evaporation would have no effect on the median diameter (assuming that condensation and evaporation have the same size dependence) and coagulation would be the primary cause of growth (Bian et al., 2017; Hodshire et al., 2019b, a; May et al., 2015; Palm et al., 2020). Here we have shown that even in cases where OAER is changing as the plume ages, coagulation is still the primary mechanism through which the diameter changes, and diameter changes due to condensation/evaporation are secondary.

The modeled results when segregated by ΔCO percentile generally show an overprediction of growth in the highest percentile bins (both with coagulation only and also when condensation/evaporation are added), and an underprediction of growth in the lowest percentile bin (Fig. 8). On average, the mean bias for the simulation without OA condensation/evaporation is larger than the typical variability of the median diameter measurements at -15 nm and 9 nm in the 5–15 and 90–100 ΔCO percentiles, respectively. The Pearson correlation coefficients between the modeled and observed $dD_{pm}/dt$ are weak (0.3) in the 5–15 ΔCO percentile and very strong (0.81) in the 90–100 ΔCO percentile (albeit with a model overprediction of growth). While the 90–100 ΔCO percentile has a similar correlation between modeled and observed $dD_{pm}/dt$ as the transect averages, the correlation in the 5–15 ΔCO percentile is weaker due to less coherent growth trends in the observations and influence from the other percentile bins. Similar to the transect averaged results, including OA condensation/evaporation based on the observed changes in OAER only changes the model agreement within the uncertainty of the measurements, and the biases remain (7 nm in the 90–100 ΔCO percentile and -19 nm in the 5–15 ΔCO percentile).

The larger magnitude of bias in the extremities of the ΔCO percentiles than that seen in the transect averages suggests that mixing between percentile regions of the plumes is occurring on a time scale slow enough that there are apparent differences between the dilute and concentrated portions of the smoke plume, but the mixing is happening too quickly for the core and edge of the plume to be treated separately when simulating aging over several hours. As described in the methods, we estimated the mixing times between the core and edge of the plume were calculated based on wind standard deviation derived stability class and Gaussian plume relations. The majority of mixing times from both the stability class and ΔCO

gradient method tend to be around 2 to 5 hours as shown in Table 2. Figure S19 shows that the two methods are strongly correlated with a Pearson correlation coefficient of 0.92. We believe that these times are supportive of the results in Fig. 8, since they are comparable to the length of time the plane may have been sampling a plume. The exception to this is the mixing times for Williams Flats 8/7 P2, which had mixing times >10 hours from both methods). Williams Flats 8/7 P2 was the only case where $D_{pm}$ in both the 5–15 and 90–100 ΔCO percentiles was simulated within the uncertainty of the measurements, which is additional evidence that in this case the mixing was slow enough that treating the percentiles as separate was a valid assumption. Vertical mixing, not captured here, may also influence results for the faster mixing cases; for example, vertical mixing in the plume on August 3 was evident in large eddy simulations (LES) of the first pass on this day (Wang et al., 2021). In the LES simulation, dilution and physical mixing strongly impacted the chemistry within the smoke plume, but that study did not examine how the mixing impacted the particle diameters in the smoke plume (Wang et al., 2021).

## 4 Conclusions

Using data from eight pseudo-Lagrangian samplings of western US wildfires during the FIREX-AQ campaign and simulations of growth using a sectional aerosol microphysics model, we examined the impact of initial OA mass concentration (ΔOA$_i$) on the observed aerosol size distribution, organic aerosol enhancement ratio (OAER), and ΔO:ΔC evolution in the first 3 to 7 h of physical smoke aging. Despite variability in the age at the first transect, we are able to use this experimentally derived starting point to determine relationships between plume concentration at the first transect and the subsequent evolution. Observations showed that relatively high-smoke-concentration plumes (ΔOA$_i$ > 1000 µg m$^{-3}$) exhibit more particle evaporation after the first transect than lower-concentration plumes (ΔOA$_i$ < 100 µg m$^{-3}$), but that this increase in evaporation is not sufficient to offset particle growth due to coagulation. Consequently, the net effect is that the high-concentration plumes have faster particle diameter growth than the lower-concentration plumes. Further, regardless of concentration we are able to simulate that coagulation explains a majority of the growth for many pseudo lagrangian transects. The rate at which number enhancement ratio decreased was not significantly correlated to ΔOA$_i$ and the model performed less strongly in predicting the number enhancement ratio than the median diameter. It is possible these discrepancies are in part due to deviations from true Lagrangian sampling and uncertainties in the LAS saturation at high concentrations. Thus, improved understanding of how the emissions changed as the smoke was being sampled due to deviations from true Lagrangian sampling, and of how the LAS saturates at high concentrations would be beneficial to improve these analyses.

At the first transect, initial OAER and initial ΔO:ΔC suggest that less concentrated plumes have faster evaporation prior to the first transect than more concentrated plumes. After the first transect, ΔO:ΔC always increased in the smoke plumes with no correlation to the plume concentration, while rates in OAER change as the plume ages vary with plume concentration such that net evaporation as the plume ages is more likely in the more concentrated plumes. Dilution-driven

evaporation is likely important in these OAER decreases and $\Delta O{:}\Delta C$ increases seen in these smoke plumes. In plumes with no significant OAER change, there is likely a balance between POA evaporation and SOA formation. Additional modeling of OA and its composition would improve understanding of the relative roles of evaporation and SOA formation in plumes of varying concentrations.

Dividing the plume into dilute and concentrated sections based on $\Delta CO$ percentiles, showed changes in diameter, number enhancement ratio, and OAER with smoke age to be dependent on $\Delta OA_i$. However, physical mixing within the plume limits the ability to simulate $\Delta CO$ percentiles independently, especially on the edges of smoke plumes, which experienced more growth than simulated. Mixing within the plume was not considered in prior use of this methodology. Hence, Lagrangian sampling of a wider range of plume concentrations, or sampling plumes under very stable conditions with limited mixing, would help to improve the understanding of how smoke plume concentration influences its evolution.

Future work includes using a dispersion-resolving model with online chemistry and aerosol microphysics schemes to better examine the results found here relating to in-plume gradients and OA evaporation/condensation. Simulations of this type would also help to better quantify vertical and horizontal mixing occurring in the smoke plumes. Additionally, continued work in understanding how the details of how spatiotemporally varying emission ratios impact the plume aging would be beneficial as our results here do not take into consideration fuel types.

**Data Availability**

All data are publicly available in the NASA FIREX-AQ data archive (http://doi.org/10.5067/SUBORBITAL/FIREXAQ2019/DATA001).

**Author contribution**

NAJ performed the analysis, contributed to the direction of the analysis, and led the writing of the paper. JRP conceived/directed the analysis and contributed to code development and writing. ALH contributed to the direction of the analysis and assisted in writing. EBW, ELW, CER, KLT, KJS, RHM, JLJ, JMD, TPB, JP provided measurements. RHM, MMC, RJY, MJA, SMK, and SHJ contributed to the direction of the analysis. All authors provided feedback on the manuscript.

**Competing interests**

The authors declare that they have no conflict of interest.

## Acknowledgements

This work is supported by the US NOAA, an Office of Science, Office of Atmospheric Chemistry, Carbon Cycle, and Climate program, under the cooperative agreement awards NA17OAR4310001 and NA17OAR4310003; the US NSF Atmospheric Chemistry program, under grant AGS-1950327; and the US Department of Energy's (DOE) Atmospheric System Research, an Office of Science, Office of Biological and Environmental Research program, under grant DE-SC0019000. This publication was also developed under Assistance Agreement No. R8400008 awarded by the U.S. Environmental Protection Agency (EPA) to Shantanu Jathar. It has not been formally reviewed by EPA. The views expressed in this document are solely those of the authors and do not necessarily reflect those of the Agency. EPA does not endorse any products or commercial services mentioned in this publication. The NASA Langley Aerosol Research Group (LARGE) was supported by the NASA Tropospheric Composition Program. RJY: NASA 80NSSC19K1589, NOAA NA16OAR4310100. We thank the pilots and crew of the NASA DC-8 and the FIREX-AQ project scientists: Jim Crawford, Jack Dibb, Carsten Warneke, Shuka Schwarz, and Barry Lefer.

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

**Table 1:** Information on the fires used in analysis. Fuels are from the National Wildfire Coordinating Group (NWCG) incident reports (https://inciweb.nwcg.gov/).

| Flight Date | Fire | Number of sets of pseudo-Lagrangian transects | Fuel | Smoke age vs. flight time slope |
|---|---|---|---|---|
| 7/25/2019 | Shady | 1 | Timber, tall grass | 2.15 |
| 7/29/2019 | North Hills | 1 | Tall grass, medium logging slash | 2.55 |
| 8/3/2019 | Williams Flats | 2 | Dead trees, grass, sage, bitterbrush | 3.97, 2.34 |
| 8/6/2019 | Williams Flats | 1 | Dead trees, grass, sage, bitterbrush | 2.88 |
| 8/7/2019 | Williams Flats | 2 | Timber, brush, short grass | 2.96, 2.94 |
| 8/12/2019 | Castle | 1 | Timber | 3.16 |

**Table 2:** The distance between the average location of the 90–100 $\Delta$CO percentile (Core) and the innermost location in the 5–15 percentile bin (Edge) and the corresponding time it takes for the plume to mix that distance.

| Flight | Core Edge Distance [m] | Stability Class Mixing Time [h] | $\Delta$CO Gradient Mixing Time [h] |
|---|---|---|---|
| 7/25 | 6291 | 3.8 | 1.7 |
| 7/29 | 2695 | 2.6 | 3.5 |
| 8/3 P1 | 6321 | 4.0 | 4.6 |
| 8/3 P2 | 11470 | 4.6 | 1.8 |
| 8/6 | 8604 | 3.1 | 1.3 |
| 8/7 P1 | 10841 | 8.9 | 2.9 |

| 8/7 P2 | 13153 | 23.4 | 12.6 |
| 8/12 | 12786 | 4.5 | 3.8 |

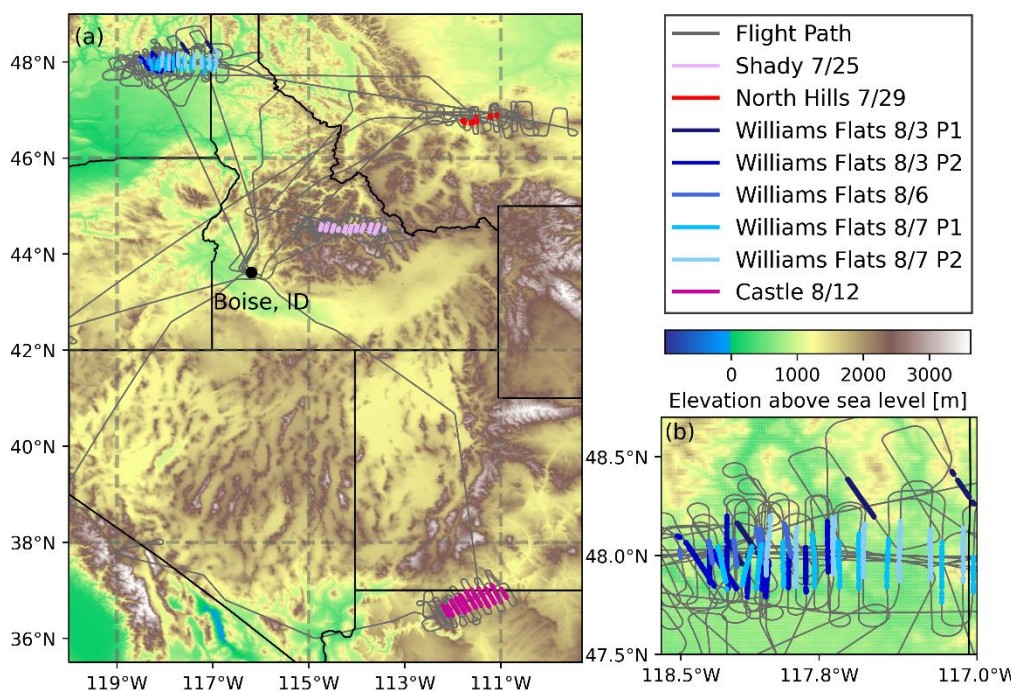

**Figure 1:** (a) Map of in-plume sections for the eight sets of transects used in this study from the FIREX-AQ campaign
between July 25 and August 12, 2019. (b) Map of the in-plume sections of the five sets of transects of the Williams Flats
Fire.

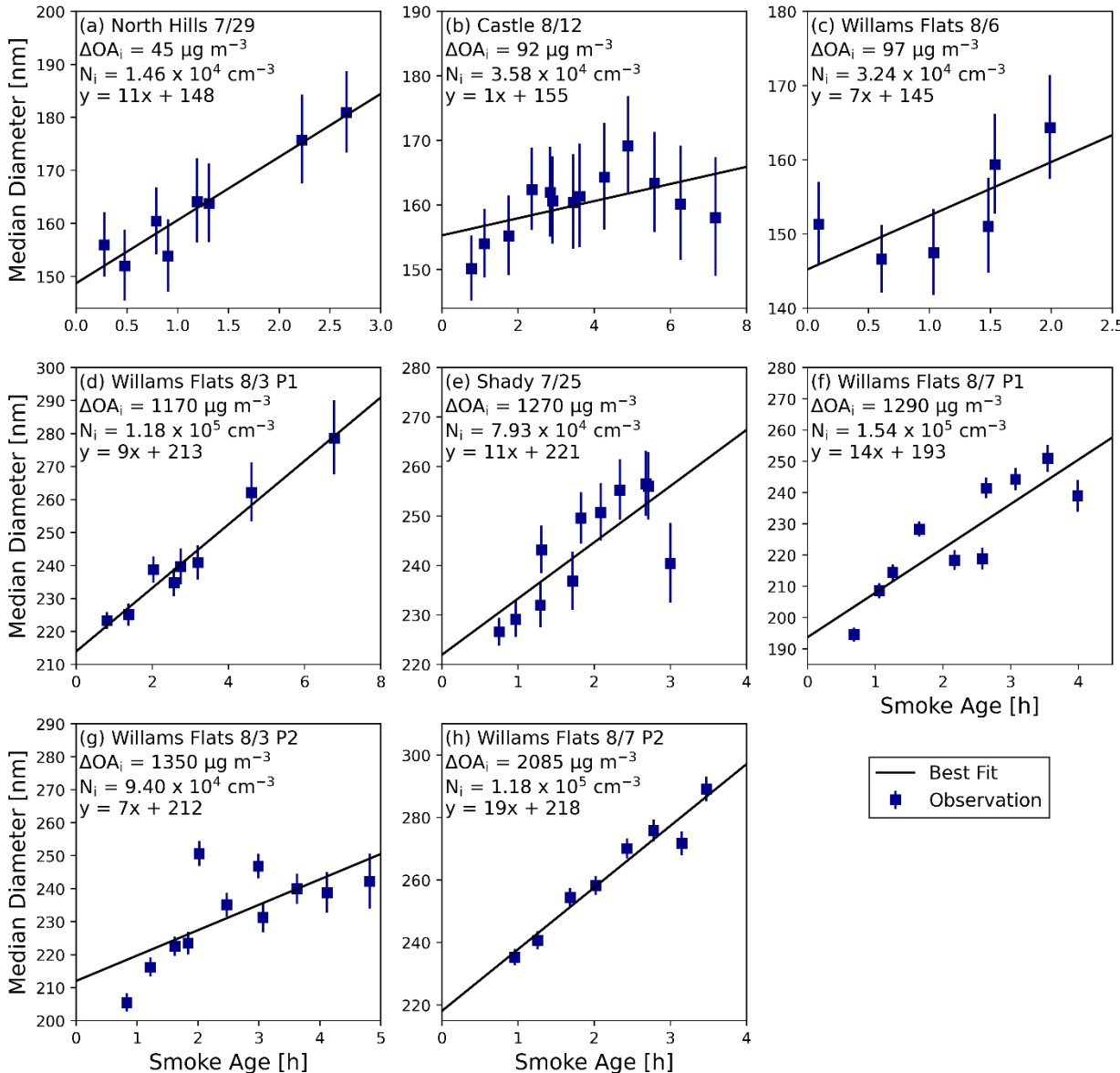

**Figure 2:** The median diameter ($D_{pm}$) versus smoke age for each of the eight flights, organized so that (a)–(h) are in order of increasing $\Delta OA_i$. The error bars represent the standard deviation of $D_{pm}$ within the transect.

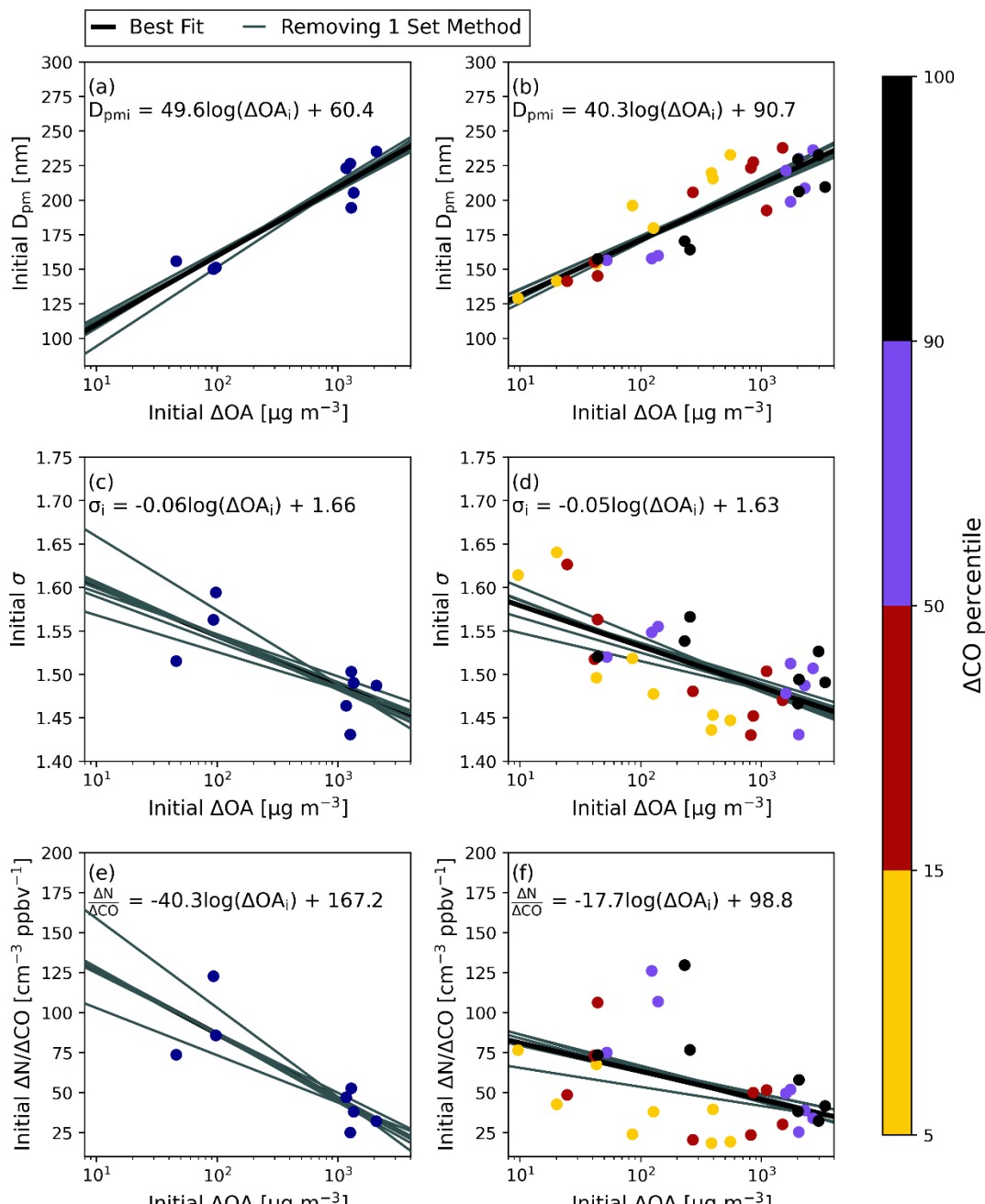

**Figure 3:** Initial $D_{pm}$ versus initial $\Delta OA_i$ in the transect averages (a) and $\Delta CO$ percentiles (b). Initial $\sigma$ versus initial $\Delta OA_i$ in
the transect averages (c) and $\Delta CO$ percentiles (d). Initial $\Delta N/\Delta CO$ versus initial $\Delta OA_i$ in the transect averages (e) and $\Delta CO$
percentiles (f). On each panel, the best fit line for the points is shown in solid black with the equation of this line shown on
the panel. The gray lines are the results of linear regressions with one set of transects removed at a time.

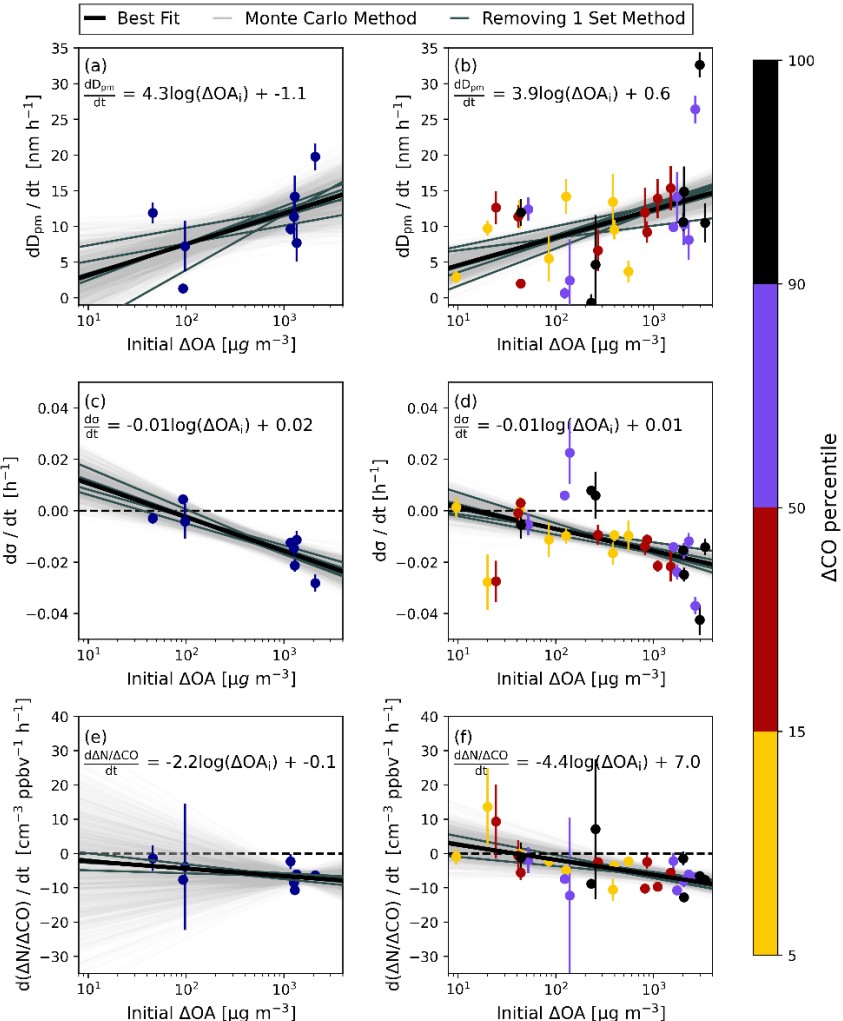

**Figure 4:** Observed average rate of change growth rate of the $D_{pm}$ (a-b), $\sigma$ (c-d), and $\Delta N/\Delta CO$ with smoke age for the 8 sets of transects based on ordinary least squares linear regressions as a function of $\log(\Delta OA_i)$ (initial background-corrected organic aerosol). Rates of change are for trends in the transect average values in (a), (c), (d) and the $\Delta CO$ (background-corrected CO) percentile ranges for each set of transects in (b), (e), (f). The error bars indicate the 95% confidence interval

for the average rates of change on the y-axis. One thousand best fit lines from a Monte Carlo technique are shown in light gray. The average and slope and intercept with their respective 95% confidence interval for the Monte Carlo fits are shown in Table S3. The darker gray lines are the results of linear regressions with one set of transects removed at a time. The solid black line is the linear regression for the points at the center of the error bars; the equation for this line is shown on each panel.

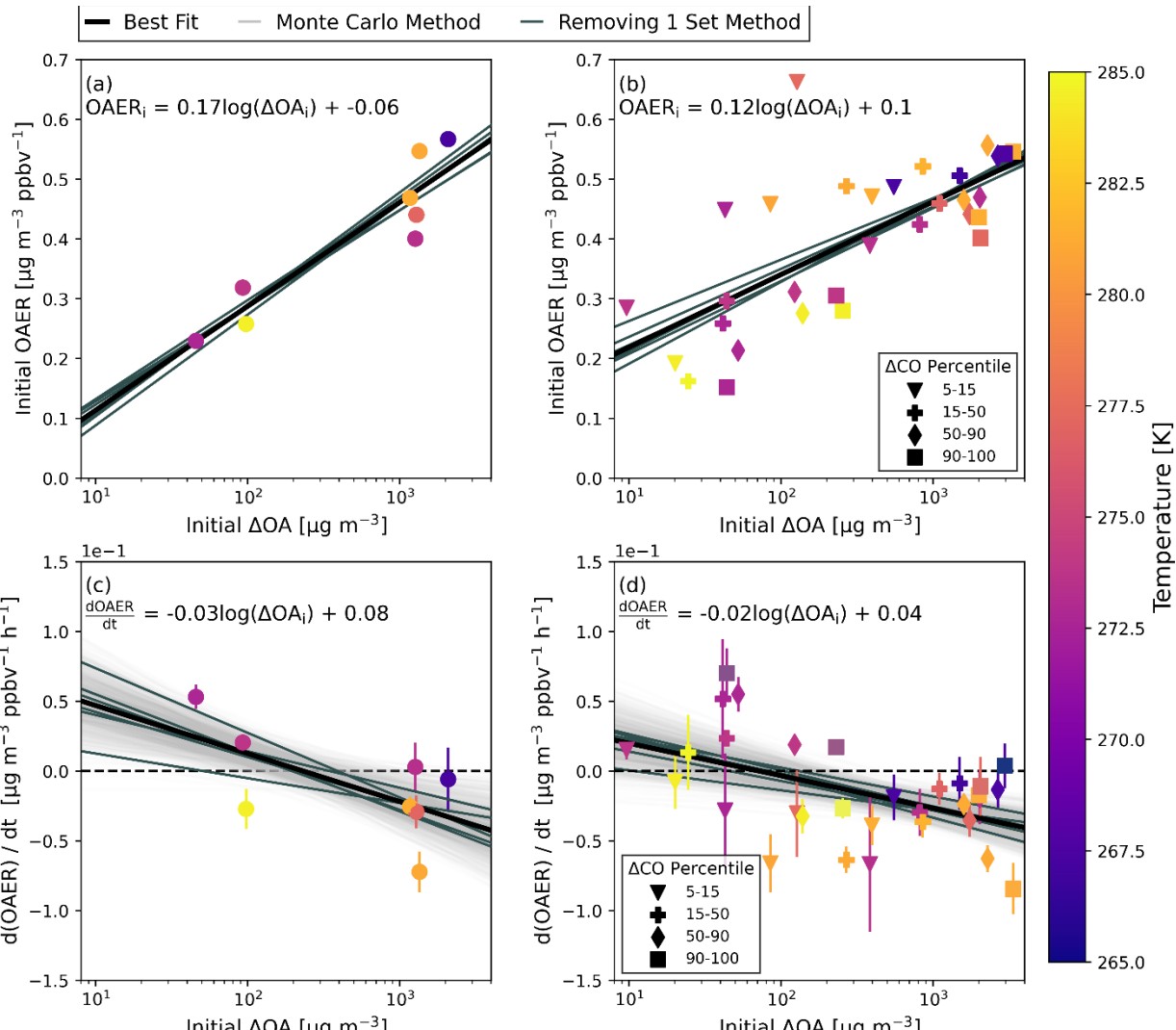

**Figure 5:** Initial OAER ($\Delta$OA/$\Delta$CO) versus initial $\Delta$OA for (a) the transect averages and (b) by $\Delta$CO percentile colored by the average in plume temperature at the first transect with an OLS regression line in gray. The statistics for this fit are shown in Table S3. The OAER trends with smoke age based on OLS fitting as a function of $\Delta$OA$_i$ for (c) the transect averages and (d) by $\Delta$CO percentile respectively. On each panel, the best fit line for the points is shown in solid black with the equation of this line shown on the panel. The darker gray lines are the results of linear regressions with one set of transects removed at a time. On (c) and (d), 1000 best fit lines from a Monte Carlo technique are also included in light gray with statistics for these fits shown in Table S3. The black dashed line on (c) and (d) is the y=0 line.

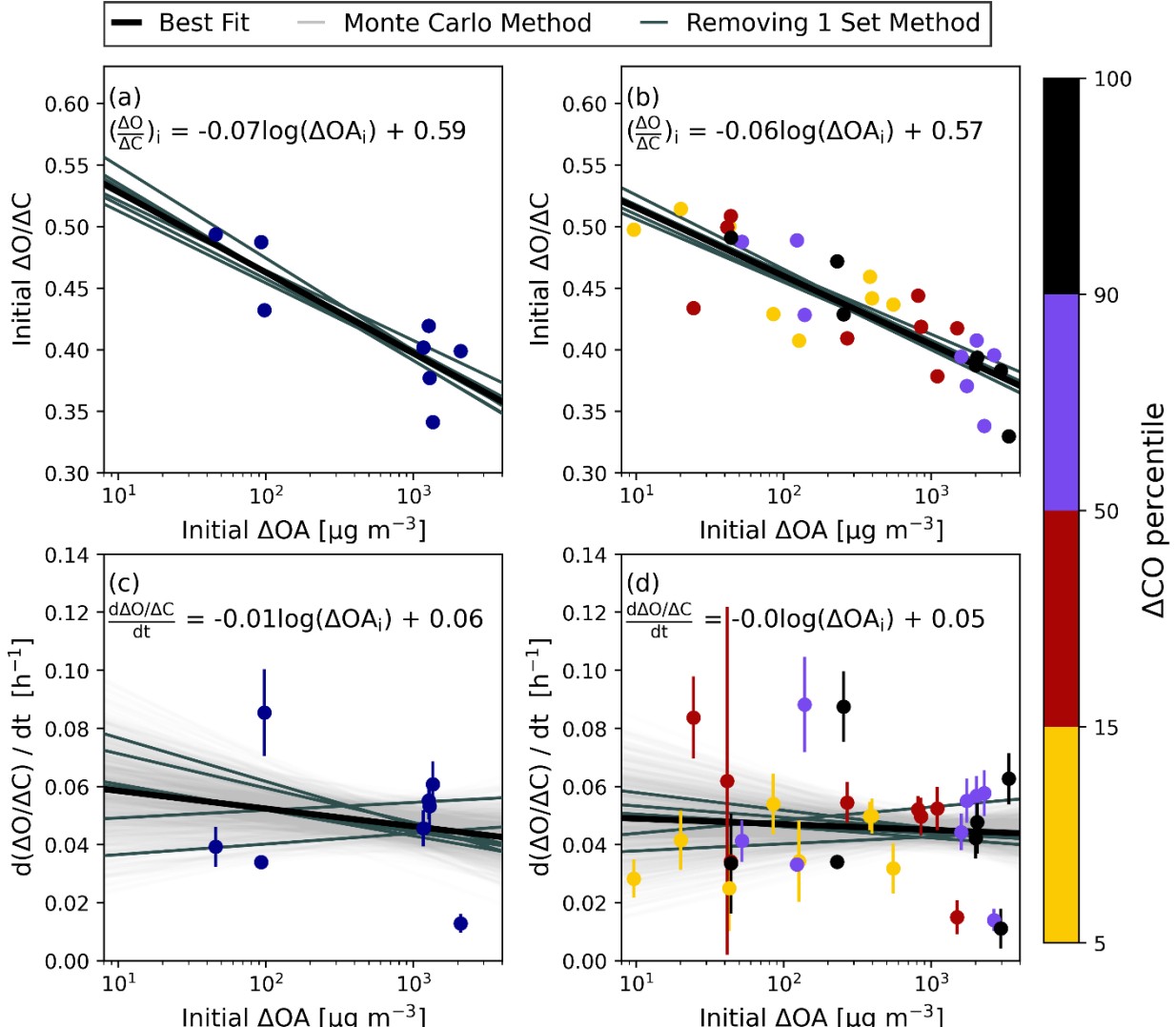

**Figure 6:** The initial ΔO:ΔC at the first transect versus initial ΔOA for (a) the transect averages and (b) by ΔCO percentile. The linear fit slopes of ΔO:ΔC with smoke age versus $\Delta OA_i$ for (c) the transect averages and (d) by ΔCO percentile. On each panel, the best fit line for the points is shown in solid black with the equation of this line shown on the panel. The darker gray lines are the results of linear regressions with one set of transects removed at a time. On (c) and (d), 1000 best fit lines from a Monte Carlo technique are also included in light gray with statistics for these fits shown in Table S3.

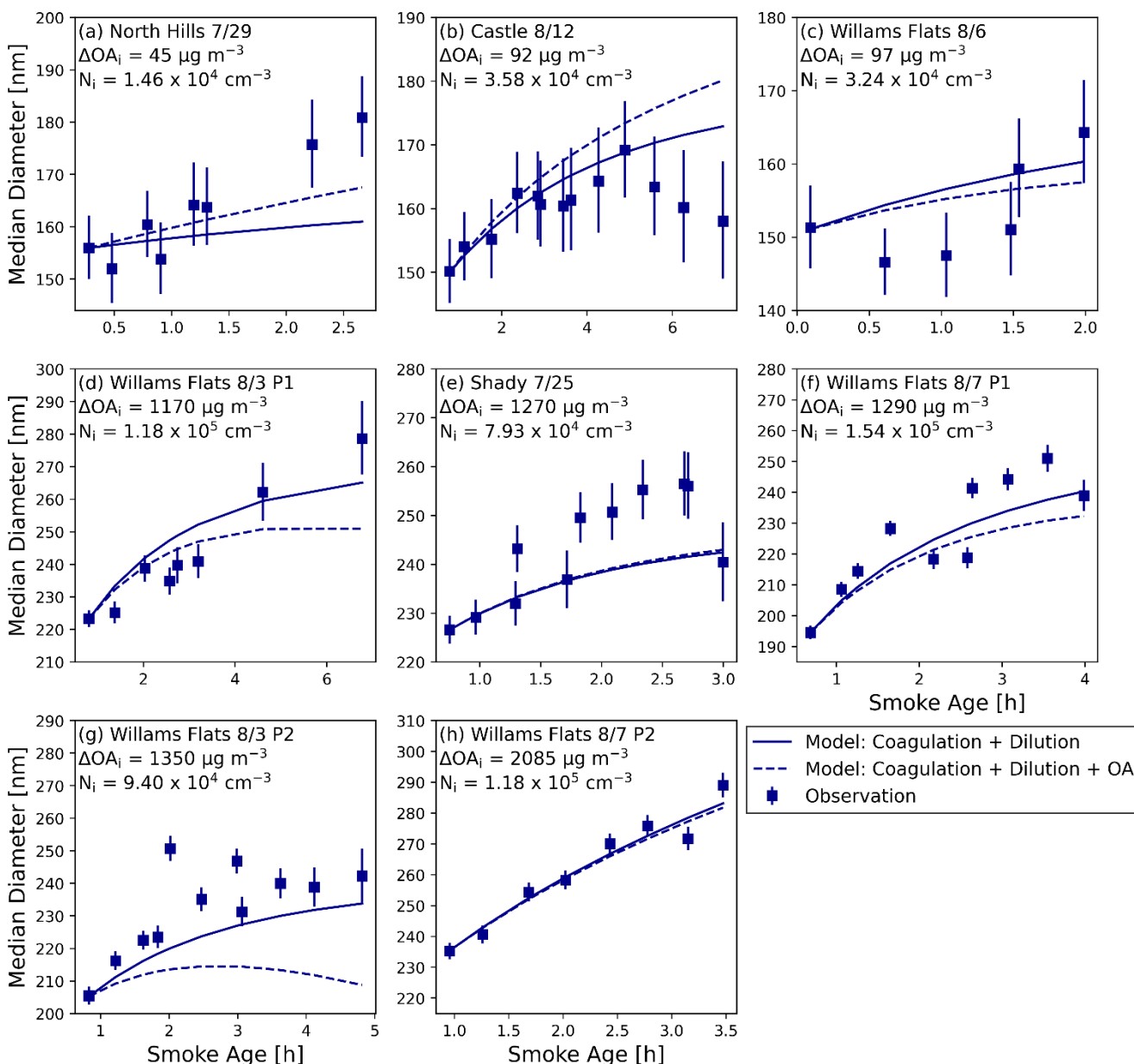

**Figure 7:** The observed median diameter ($D_{pm}$) (points), modeled $D_{pm}$ due to coagulation and dilution alone (solid line), and modeled $D_{pm}$ due to coagulation and dilution plus diameter changes due to OA evaporation/condensation (dashed line) as a function of smoke age for each of the eight smoke plumes used in our analysis. The error bars represent the standard deviation of $D_{pm}$ within the transect. On each panel is $\Delta OA_i$, and the aerosol number concentration of particles between 50 nm and 800 nm measured at the first transect ($N_i$). (a)–(h) are in order of increasing $\Delta OA_i$.

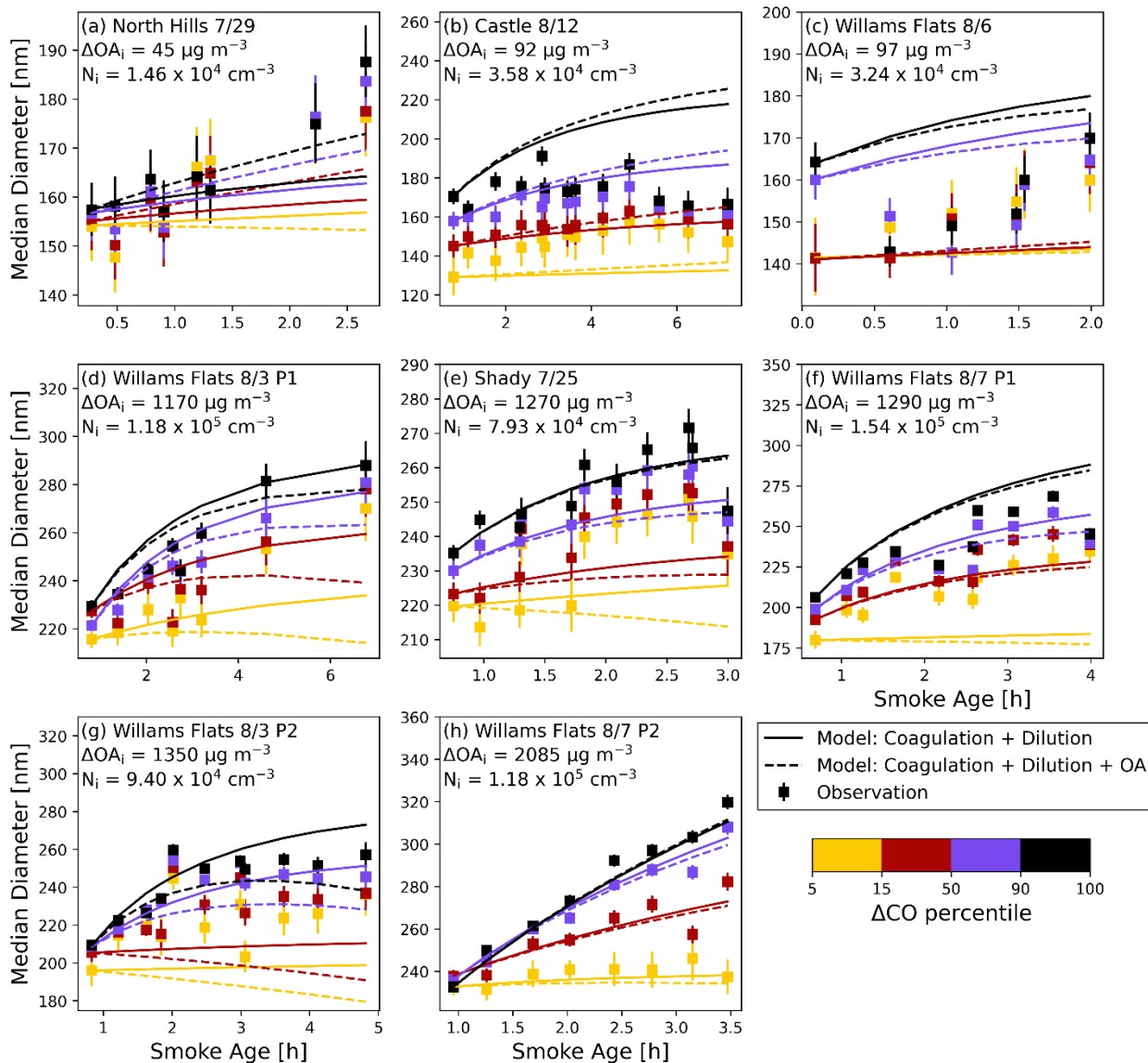

**Figure 8:** The observed (points), coagulation modeled (solid line), and coagulation plus changes due to OA evaporation/condensation (dashed line) median diameter as a function of smoke age for each of the eight smoke plumes used in our analysis colored by $\Delta CO$ percentile. Shown inset is the $\Delta OA$ measured at the first transect ($\Delta OA_i$) in µg m$^{-3}$, and the aerosol number concentration of particles greater than 100 nm measured at the first transect ($N_i$) in # cm$^{-3}$. The error bars represent the standard deviation of $D_p$ within the transect. (a)–(h) are in order of increasing initial $\Delta OA$.