# Peer review of "Aerosol size distribution changes in FIREX-AQ biomass burning plumes: the impact of plume concentration on coagulation and OA condensation/evaporation"

_Atmospheric Chemistry and Physics, 2022_

## Author Comment (AC1)

**Referee #1**

We thank the referee for their time and helpful comments.

Major comments

Key factor controlling aerosol size distribution in biomass burning plumes is interesting and important topic in current air quality and climate change studies. This work analyzed the impact of plume concentration on coagulation and OA condensation/evaporation based on FIREX-AQ aircraft measurements. They found the median particle diameter increases faster in smoke of a higher initial OA concentration than smoke of a lower initial OA concentration. Their box model simulation suggested that coagulation explains the majority of the diameter growth and OA evaporation/condensation having a relatively minor impact on the diameter growth. Although the authors provided systemic analysis with FIREX-AQ measurements, the results found by them are not novelty. Similar results have been reported by previous studies. The authors need to highlight their new findings in this work.

*There are several novel aspects of our study, and we have now added text to point these out more explicitly. While previous studies have concluded that coagulation explains the majority of aerosol number size distribution changes, we're unaware of a study that has investigated this for a large number of pseudo-Lagrangian samples. We show in field data that coagulation dominates the aerosol size distribution changes, this had only been speculated in prior field data and shown in theoretical studies. We also investigate in-plume gradients in much greater detail than the Hodshire et al. (2021) study, showing relationships between plume concentration and plume evolution in both transect averaged and in gradients within the plume. We show through modeling and calculated mixing times that the mixing within the plume is generally too fast to be able to apply the relationships derived from in-plume gradients to plumes of a similar average concentration. Mixing was not considered in the prior Hodshire study investigating in-plume gradients in this way. The limitations of prior studies are discussed in the introduction. We have added to the final paragraph of the introduction lines stating what is new when we overview the work of this manuscript.*

*"We use an aerosol-microphysics model to estimate how much of the aerosol size growth is due to coagulation versus OA condensation/evaporation; the first study to show in multiple Pseudo-lagrangian transects of smoke plumes the dominance of coagulation. Finally, we investigate the timescale of mixing between the more and less concentrated regions of plumes to determine if aging in these portions of the plumes can be assumed to occur independently; prior studies have not investigated this role of mixing."*

For the writing style of this manuscript, too much Figures were put in supporting material with unnecessary. It broke the logical flow of the paper.

*We (at least June and Pierce, the main 2 authors writing the manuscript and responding to the comments) have tried to balance relative conciseness (still 8 figures and 2 tables in the revised manuscript, so not \*that\* concise) and completeness (wanting to show correction*

*factors, sensitivities to assumptions, and the underlying data). We feel that moving figures from the SI to the main text will make the paper unwieldy for the average reader, while deleting SI figures would create a "just take our word for it" aspect to the paper that we seek to avoid.*

There are some scientific questions need to be addressed.

1. Most of the authors' analysis is based on smoke age. However, the definition of smoke age in this work is too simple and seems not to reflect the impact of turbulence and wind shifting. However, the uncertainties of smoke age could impact significantly on the authors' analysis, discussions, and conclusions.

*We have added a sentence acknowledging the uncertainties related to smoke age. We also now discuss how we believe these are likely smaller than other uncertainties in the data (e.g., the known issues of imperfect Lagrangian sampling). We continue to use the straight line horizontal advection definition of smoke age, as this is how the physical smoke age has been defined in prior studies examining the physical aging of smoke.*

*"Although there are likely uncertainties in the smoke age due to the wind shifting directions and wind velocity varying within the plume, these uncertainties are likely smaller than the uncertainty due to issues of imperfect Lagrangian aircraft sampling."*

2.      One of the major conclusions is the authors found OA evaporation/condensation having a relatively minor impact on the diameter growth. However, this conclusion is based on box model simulation. As shown in the simulations presented in the paper, OA condensation in young age plume and OA evaporation in old age plume seem not to be reflected by the box model.

*The OA changes are not predicted by the model. Rather, OA evaporation/condensation in the model is given as an input from the observed changes in the OA enhancement ratio. This is detailed in the methods (Sect. 2.2), and stated with the modeled results "OA condensation/evaporation was included in the model based on the observed trends in OA (Fig. 5c) and Eq. 6." We are unclear what time scales are being referred to as young and old in this comment; however, past observational studies have shown that in near term aging 0 to 8 hours, as we consider in this study, OA can experience either condensation or evaporation or no net change. In the theoretical studies of Bian et al. (2017) and Hodshire et al. (2019b), quickly diluting plumes were predicted to have net evaporation in the first ~30 minutes followed by net condensation (the opposite that the reviewer suggests), and the initial OA:CO ratios tend to support this, where the plumes with lower OA concentrations at the first transect have lower OA:CO ratios at the first transect, which is evidence of early initial evaporation in these plumes. The plume simulations, however, start at the first transect and hence do not include this evaporation before the first transect.*

3.      The authors used linear regression to fit the dependence of median diameter with smoke age. Median diameter cannot increase without limitation. I do not think linear regression is suitable to be used here.

*We added a sentence to the methods section noting that the linear regression slopes are not intended to be extrapolated beyond the time frame of the FIREX-AQ observations (2 to 7 hours). We refer to the slope from the linear regression of median diameter with smoke age as an average growth rate, which suggests the instantaneous growth rate is not necessarily constant, even within the observed time period. We chose the use of linear regression since it allows us to determine the relationship of a single parameter (average growth rate) with plume concentration. The use of a non-linear function would introduce multiple parameters, limiting the ability to clearly determine the impact of plume concentration on growth. We agree that median diameter cannot increase without limitation, this is discussed in the paper in the modeling section, where we point out differences in the dilution rate leading to a faster/slower decrease in the coagulation rate.*

Minor comments

Line29: μg m-3. -3 needs to be superscript. Please check the unit in the paper.

*Fixed.*

Line200: Change x to multiple sign.

*Fixed.*

Line 239-240: How do the authors think about the impact of wind direction/shifting on their method to calculate the smoke age? Wind direction may not always follow the straight-line between the fire and aircraft position.

*As noted above, we have added a sentence acknowledging the uncertainty in the smoke age due to assuming straight-line advection.*

*"Although there are likely uncertainties in the smoke age due to the wind shifting directions and wind velocity varying within the plume, these uncertainties are likely smaller than the uncertainty due to issues of imperfect Lagrangian aircraft sampling."*

Page8Equation2: It is unclear what do inplume and background refer to. Does inplume refer to the value sampled at aircraft position in plume? Where are the background values sampled?

*We have added definitions of the "in plume" and background subscripts used in the equation to the prior paragraph. And we have added a sentence on where background values are sampled.*

*"The concentration enhancement of species X due to the presence of smoke ($\Delta X$) is determined by subtracting the average background concentration ($X_{background}$) of this species from the in-plume measurements ($X_{inplume}$). Background concentrations are an average concentration measured outside the plume at the same altitude as the aircraft sampled the plume."*

Line304: More details about the assumptions, settings, and initial conditions to run the coagulation model are needed. Based on current information, it is hard to reproduce the results presented in this work.

*We have added details about the temperature and pressure used in coagulation kernel calculation, and we have rearranged the paragraph so the initialization information is easier to find.*

*"In the Brownian coagulation kernel calculation, we assume a particle density of 1400 kg m$^{-3}$, and assume the temperature and pressure are the average of the in-plume measurements."*

*In the previous version, standard temperature and pressure were used in our simulations with number concentration in standard cm$^{-3}$. In this version, we have adjusted the simulations to use ambient number concentration and average temperature and pressure of the in-plume measurements. The simulation results are similar to the previous version and conclusions have not changed.*

Line339-341: I agree with the authors' discussion that the diameter growth rates should be slowed with the smoke age. However, according to Fig.2, we did not see such slowdown. The authors need to explain this disagreement.

*We have added a sentence noting the variability of the growth rate slowing between transects. We discuss this in context of the model results shown in Fig. 7.*

*"The rate of the growth slow down varies between sets of transects, and in days such as Williams Flats 8/7 P2 the slow down is not noticeable to slow dilution; the growth slow down is discussed more in Sect. 3.2 with the model results."*

Line455: Why does median diameter drop after 5h in Fig.7b? Why is median diameter low during 0.5-1.5h and then rapid increased after 1.5h in Fig.7c?

*For both of these, we believe they are due to imperfect Langrangian sampling by the aircraft (both in terms of rate of flying downwind and potentially also the vertical position within the plume), which results in variability in the observed median diameter. The observed*

*median diameter is discussed with Fig. 2 and Fig. 3 in Sect. 3.1. We have added a sentence to briefly restate these claims in the discussion of Fig. 7.*

*"Some of the disagreement between the model and observations may be due to the lack of pseudo-Lagrangian aircraft sampling, especially noticeable in Fig. 7b and 7c, as discussed in Sect. 3.1."*

Line475-477: Need to explain the reason why the change from non-size-dependent evaporation to size-dependent evaporation in model is small.

*This change is small because the assumed diameter (300 nm) used in the non-size dependent evaporation correction is near that of the observed peak diameter. We have added text to clarify.*

*"Changing from the non-size-dependent evaporation correction to a size-dependent evaporation correction based on Fig. S4b does not change the agreement of the modeled and observed median diameters because the modeled coagulation rate is unchanged, and only a minor shift in median diameter occurred because the assumed diameter of 300 nm used for the non-size-dependent evaporation correction is near the observed peak diameter (Fig. S14)."*

Line478: Can it be caused by the treatment of dilution in the box model?

*The sentence at this line is, "The observed trends in the number enhancement ratio are noisier than the trends in $D_{pm}$, but the model still is able to capture some of the reduction in number as a result of coagulation (Fig. S13)." This sentence is primarily about noise in the observations, so we are unsure what the reviewer is suggesting. We cannot think about why the treatment of dilution in the box model would help the model capture the noise in the number enhancement observations.*

Line489-490: Why does the impact of changing emission on number concentration is larger than size distribution?

*Our hypothesis is that the N:CO emissions ratio may change with fire changes over several hours while the emitted diameter is not changing. We have added this as a discussion point related to Fig. 4.*

*"The aerosol number enhancement ratio could be less correlated with smoke age than Dpm due to a changing N:CO emissions ratio from the fire during the period of imperfect pseudo-Lagrangian sampling (with the plane moving downwind ~4x faster than the wind speed)."*

Line491-492: In Fig.7b and Fig.7g, the consideration of OA evaporation/condensation reduced the agreement of simulation and observation. More discussions on this are needed. Usually, young age plume is dominated by OA condensation, while old age plume is dominated by OA evaporation. I did not see this reflected in these simulations. Do the authors know why the simulation works this way?

*We have added discussion to the text on the reduction of agreement between the modeled and observed median diameter for two of the cases.*

*"Two cases, Castle 8/12 and Williams Flats 8/3 P2, had reductions in model agreement with the inclusion of OA condensation/evaporation. For Castle 8/12, condensation resulted in an overestimation of the growth; however, this bias is not greater than the uncertainty of the measurement. Net evaporation was observed during Williams Flats 8/3 P2 resulting in underprediction of the growth."*

*As noted in the text, the OA evaporation/condensation in the model is constrained by the observations (Fig. 5c and Fig. S8). The observations do not show condensation in the early transects with evaporation in the older transects. Our study is constrained to the 0 to 8 hours of aging that were sampled by the aircraft.*

Page28Fig1: Map shown here is helpless to readers who are not familiar with local geography of Idaho. Map with land use type is more useful.

*We have added elevation of the region to the map.*

---

## Author Comment (AC2)

**Referee #2**

We thank the referee for their time and helpful comments.

This manuscript presents airborne observations of some aerosol parameters (aerosol particle median diameter, number concentration of particles > 100 nm, organic aerosol enhancement ratio (OAER, deltaOA/deltaCO) and O:C ratio) as function of atmospheric age for eight major wildfire plumes in the western US in 2019. The main finding of the manuscript is that the median diameter of particles > 100 nm increases faster in plumes that have higher OA concentration. A box model suggests that this change is mainly due to coagulation, OA evaporation and condensation having a minor role.

Biomass burning aerosol size distribution and its evolution in sub-grid scale is an important topic for climate and air quality. However, it is unclear what are the novel findings in this study and there are also some major comments need to be addressed before this manuscript can be accepted.

*There are several novel aspects of our study, and we have now added text to point these out more explicitly. While previous studies have concluded that coagulation explains the majority aerosol number size distribution changes, we're unaware of a study that has investigated this for a large number of pseudo-Lagrangian samples. We show in field data that coagulation dominates the aerosol size distribution changes, this had only been speculated in prior field data and shown in theoretical studies. We also investigate in-plume gradients in much greater detail than the Hodshire et al. (2021) study, showing relationships between plume concentration and plume evolution in both transect averaged and in gradients within the plume. We show through modeling and calculated mixing times that the mixing within the plume is generally too fast to be able to apply the relationships derived from in-plume gradients to plumes of a similar average concentration. Mixing was not considered in the prior Hodshire study investigating in-plume gradients in this way. The limitations of prior studies are discussed in the introduction. We have added to the final paragraph of the introduction lines stating what is new when we overview the work of this manuscript.*

*"We use an aerosol-microphysics model to estimate how much of the aerosol size growth is due to coagulation versus OA condensation/evaporation; the first study to show in multiple Pseudo-lagrangian transects of smoke plumes the dominance of coagulation. Finally, we investigate the timescale of mixing between the more and less concentrated regions of plumes to determine if aging in these portions of the plumes can be assumed to occur independently; prior studies have not investigated this role of mixing."*

Major comments

The evolution of the size distribution is described through two parameters only: median diameter and number concentration. Furthermore, the size distribution is available only for particles > 100 nm in diameter - were there really no measurements below 100 nm? Even for particles > 100 nm, please show the actual size distributions. Does the initial size distribution depend on OA concentration? Does the width of the size distribution agree with the modelled time evolution? Is one mode enough to describe the size distribution or should more modes be used?

*FIREX-AQ did have a SMPS on the aircraft to measure particles smaller than 100 nm. However, the SMPS scans take 60 s, during which the aircraft traveled approximately 8 km, this makes it too slow to use in our analysis. Moore et al. (2021) showed that for points when the SMPS was scanning the plume, the LAS captured the majority of the mode, and there was not a clear, smaller Aitken mode.*

*We have added a SI Figure (Fig. S5) to show the measured size distributions along with the lognormal fits of the size distribution for every transect for each of the eight cases. We believe that based on the shown fits of the LAS binned size distribution, that one mode is enough to describe the size distribution (as well as the findings in Moore et al. (2021)). Additionally, the use of the fit allows us to estimate particles in the tail of the distribution below 100 nm.*

*Investigating modal width is an excellent suggestion. We have changed Fig. 3 to now be a 6 panel plot showing the initial diameter, width, and number enhancement ratio as a function of OA concentration for the transect averages and ΔCO percentiles. All of these variables are related to the initial OA concentration, we have added a paragraph discussing this. And Fig. 4 is now the average rate of change of median diameter, modal width, and number enhancement ratio as a function of OA concentration in both the transect averages and ΔCO percentiles. We have added discussion on the rate of change of modal width.*

*We have added an SI Figure (Fig. S16) that showing the comparison between modeled and observed width. The model captures some, but not all of the reduction in the width. As well, Fig. S17 has been added to show the normalized initial and final size distributions in the model and observations.*

It seems that most deviations from the modelled time evolution (e.g. Fig. 7) are explained by sampling not being Lagrangian. Please try to identify the processes that cause these deviations and discuss them in more detail. Can the model be used to evaluate whether changes in dilution and photochemical age would be enough to explain the observed differences, or if the differences are due to other factors (e.g. emission changes)?

*In our discussion, we have added, "Non-Lagrangian sampling may be impacting the observations through a plume injection height change, so the aircraft is no longer sampling the same vertical location of the plume; or an emissions factor or fire radiative power change due to the diurnal cycle of fires."*

*In its current state, the model cannot be used to evaluate whether time changes in dilution, age, and emissions would be enough to explain the observed differences. It would also be difficult to do this without a true Lagrangian set of transects to compare to. In our results section, we suggest ideas for future field studies to help to understand the impact of non-Lagrangian sampling.*

The coagulation parameterisation by Sakamoto et al. (2016) is mentioned (e.g. lines 85-92), but no comparison is made. Please evaluate the Sakamoto et al. (2016) parameterisation with your observations.

*The parameterization is difficult to initialize for the flights since it is based on a mass flux\*fire area/wind speed/mixing depth (dM/dxdz), which includes emission rates that we do not have. We estimate the dM/dxdz value by integrating the particulate matter concentrations across each transect; however, due to the aircraft attempting to sample the plume generally near its most concentrated point vertically, the estimated value is likely biased high relative to the average value in the plume. As shown in the figure below, this results in the Sakamoto parameterization being biased high. However, the Sakamoto parameterization does show more growth in the concentrated plumes, so qualitatively it follows expectations. We do not add the Sakamoto results to the main text or SI, since we are unable to represent dM/dxdz in the way that was intended by the parameterization.*

[Figure]

Minor comments

On line 254 "N" is used for nitrogen, which could be confusing as it is used for number concentration lines 251 and 257.

*Wrote out Nitrogen instead of using N.*

line 258 "N is the number concentration between 50 nm and 800 nm, the range of diameters used to fit the dN/dlogDp measurements." So are you showing only the fitted N, and not the directly measured parameter? Please use the measured quantity in the figures.

*As noted in the text, we use the fitted quantity to capture the tails of the size distribution. We have continued to use this fitted value in initializing the model, and in Fig. 4, Fig. S9 (Formerly S7), and Fig. S15 (Formerly S13). The measured quantity of each bin can be seen in the new supplemental figure Fig. S5, showing the full size distributions with age for each case. We also have expanded the range of the fit to be between 50 nm and 2000 nm.*

line 262 "number enhancement ratio" defined for the 2nd time.

*We have removed this second definition.*

line 293 I believe the Pasquill stability classes are defined for boundary layer, including convective mixing. Can this method be applied to mixing above boundary layer, where the measurements were done (line 174-175)?

*We are extrapolating to above the boundary layer in this procedure, and we now explicitly state this in manuscript, "As a check on the mixing time calculated from the stability class, since we are extrapolating the Pasquill stability class to above the planetary boundary layer, we also calculate a mixing time from the rate of change of the ΔCO gradient between the core and edge regions." Also we estimated the stability class two ways, but the second way was only discussed briefly in the main text with results in this SI. The alternate way to estimate the mixing time uses the rate of change of the CO gradient between the core and edge regions of the plume. These two methods yielded similar results, and both showed that the majority of the plumes mixed in <5 hours with a single plume (Williams Flats 8/7 P2) taking >10 hours to mix. This general agreement provides confidence in our conclusion that most of the plumes are mixing on timescales similar to the sampled aging time with the exception of Williams Flats 8/7 P2. We have added a column to Table 2 to include the mixing times derived from the CO-gradient method, as well as more discussion related to the CO-gradient method.*

line 322-323 "Additionally, this assumes volume-controlled growth/shrinkage, where all particle sizes grow/shrink by the same fractional amount, preserving the lognormal modal width." Examining the initial size distribution as function of OA concentration could help validate this assumption. At least for the median diameter (Fig. 2), there seems to be a clear distinction between more diluted plumes (Dpm ~150nm) and the more concentrated plumes (Dpm ~200nm).

*Thanks for this suggestion. We now include initial size distribution properties as Fig. 3 that show a clear relationship between the initial diameter and initial OA. In our discussion of the initial OAER values in Fig. 5, we now discuss that the smaller diameters in diluted plumes can partially be explained by the lower OAER in the dilute plumes (with coagulation prior to the first transect likely explaining the rest), with both indicating faster evaporation prior to the first measurement in the dilute plumes.*

"The lower initial $D_{pm}$ in dilute plumes (Fig. 3a) also suggests faster evaporation. Between a $\Delta OA_i$ of 100 µg m$^{-3}$ and 1000 µg m$^{-3}$, initial OAER decreases by a factor of about 0.62, which if only evaporation was occuring would suggest the particles in the 100 µg m$^{-3}$ plume would be 0.85 times smaller assuming the emitted diameter is not correlated with concentration at the first transect. We observe particles that are 0.76 times smaller at a $\Delta OA_i$ of 100 µg m$^{-3}$ compared to 1000 µg m$^{-3}$ (Fig. 3a), suggesting that evaporation prior to the first transect is contributing to smaller particle sizes for less-concentrated plumes."

line 341-343 "Castle 8/12 also has a larger uncertainty range due to a constant increase in Dpm for the first five hours of aging, but then a decrease in Dpm during the final three transects, potentially due to deviation from Lagrangian sampling." Can you use the size distributions to rule out growth of < 100 nm particles into the sizes where they can observed?

*Based on the size distributions shown in Fig. S5, there does not appear to be an increase in the number of particles in the smallest size bins. Therefore, it is unlikely that particles smaller than 100 nm are growing into the sizes where they can be observed.*

line 428 Do you mean Fig. 6a-b?

*No, this reference is meant to be referring to the initial OAER figure, which is Fig. 5a-b in the original manuscript on ACPD.*

Figure S2 Please check x-axis label.

*Fixed x-axis label to be Smoke Age [h].*

---

## Author Response (AR2)

**Referee #1**

This work used aircraft measurement and box model simulation investigated the impact of plume concentration on coagulation and OA condensation/evaporation and the associated impact on particle growth during FIREX-AQ. The authors emphasized that one of the highlights of this work is using a large number of pseudo-Lagrangian samples to investigate this topic. However, uncertainties of pseudo-Lagrangian samples were not well investigated. Unexpected performance of field data and disagreement between observation and simulation were attributed to pseudo-Lagrangian samples which weaken the conclusions made by the authors.

For major question 1, I am still not be convinced by the explanation on smoke age. Although previous studies used the straight line horizontal advection definition of smoke age in their works, they did not investigate in-plume gradients in much greater detail as that done by the authors in this work. I would like to see more discussions on why 'these uncertainties are likely smaller than the uncertainty due to issues of imperfect Lagrangian aircraft sampling'.

*We have modified the sentence quoted just above by the reviewer to reflect less confidence in the relative uncertainties:* *"In addition to the uncertainties from the pseudo-Lagrangian sampling, there are likely uncertainties in the smoke age due to the wind shifting directions and potentially wind velocity varying radially within the plume (discussed more later in this section)."*

*Additionally, we have tested how much the calculation of age may vary between the percentile bins. In the methods paragraph detailing the splitting of the transect into percentile bins to examine in-plume gradients we have added discussion on these calculations as well as an SI figure (Figure S7).*
*"The mixing may also mean that there are differences in the smoke age in the percentile bins due to the time for the initial momentum of the smoke plume to equilibrate with the velocity of the environmental air at the injection level. In Figure S7, we show the ages of each percentile bin for each transect derived separately using the mean wind speeds in the percentile bins and the distance from the fire. While the derived ages vary by around 20 to 30 minutes between the 5 to 15 and 90 to 100 percentile bins, there are no systematic differences with one bin being generally younger or older than the other. Further, the difference in the plane speed and wind speeds cause the imperfection in Lagrangian sampling to be larger than the variability in the smoke age in the percentiles. Therefore, we use the single value of smoke age for each transect included in the dataset for both the transect average and percentile bins."*

[Figure]

**Figure S7:** *The physical smoke age in the percentiles versus the sampling time since the first transect in seconds for each of the eight sets of transects. The gray lines have slopes of 4, 3, 2 and 1, with the 1:1 line representing the ideal slope for Lagrangian sampling.*

For major question 2, the authors misread my question. I did not say OA changes are predicted by the model. I meant the size changes due to coagulation, dilution and adjust for OA condensation/evaporation were simulated by the box model. The authors found the changes of the simulated diameter growth by OA condensation/evaporation (Line499-501) were small, so they concluded that OA evaporation/condensation having a relatively minor impact on the diameter growth. It needs to be noted that this conclusion is based on box model simulation done in this work. Therefore, it is worth checking the representation of aerosol microphysics processes by this box model based on information presented in the paper. As shown in Fig.7a and b, simulated diameters with adjustment for OA condensation/evaporation are larger than those without the adjustment. It means the adjustment is dominated by OA condensation. By contrast, as shown in Fig.7c, d, f and g, simulated diameters with adjustment for OA condensation/evaporation are smaller than those without the adjustment. It means the adjustment is dominated by OA evaporation. These are agreed with the information presented

in Fig.S8. Things puzzled me is that I thought the domination of OA condensation/evaporation is changing with smoke age as shown by dots in Fig.S8. But I can only see continues OA condensation in Fig.7a and b and continues OA evaporation in Fig.7c, d, f and g. The authors need to clarify why model simulation does not reflect the variance of OA condensation/evaporation with smoke age shown in Fig.S8.

*We believe that the reviewer is confused by the method we use for specifying the Dp change due to OAER changes in the model. We state in the manuscript , "OA condensation/evaporation was included in the model based on the observed **trends** in OA (Fig. 5c) and Eq. 6.", rather than including the exact OAER for every transect. To make this point more clear we have added the following sentence, "We are basing OA condensation/evaporation on the linear fit of the points in Fig. S11, and there does not appear to be any systematic change in the slope as any of the plumes age." Note that Fig. S11 in that quotation is the Fig. S8 referred to by the referee. We edit the next sentence to reference the main text figure for OAER trends "Consistent with the OAER trends (Fig. 5c), net condensation grows the particles in comparison to the coagulation-only model diameter in 3 cases, and net evaporation shrinks the model particle diameter in 5 cases (Fig. 7)." Additionally, we edit the methods section following Eq. 6 to say "Where d(OA/CO)/dt is the average observed change in the OA enhancement ratio with time from an ordinary least squares regression, and t is the simulation time." This clarification in the methods section is to emphasize we are using the average rate of change of OAER to constrain condensation/evaporation, not the difference in OAER from one transect to the next.*